# GeoSVR: Taming Sparse Voxels for Geometrically Accurate Surface Reconstruction

Jiahe Li[1]  Jiawei Zhang[1]  Youmin Zhang[2]  Xiao Bai[1,✉]  Jin Zheng[1,3,✉]  Xiaohan Yu[4]  Lin Gu[5]

[1]School of Computer Science and Engineering, State Key Laboratory of Complex Critical
Software Environment, Jiangxi Research Institute, Beihang University
[2]Rawmantic AI   [3]State Key Laboratory of Virtual Reality Technology and Systems, Beijing
[4]Macquarie University      [5]Tohoku University
{lijiahe, baixiao, jinzheng}@buaa.edu.cn

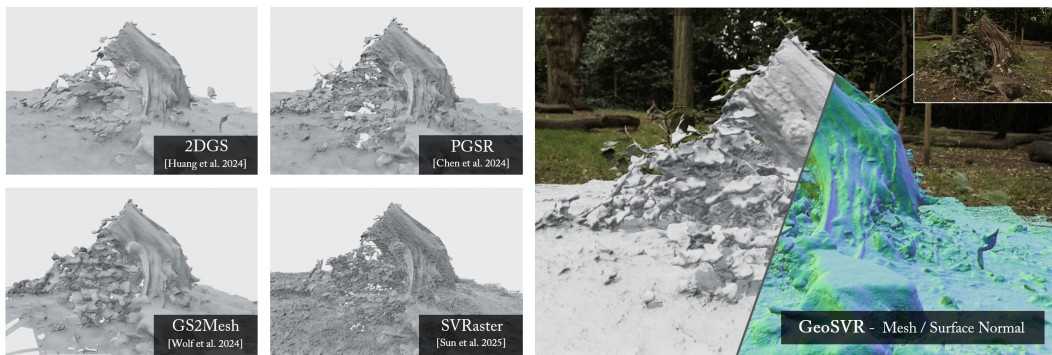

Figure 1: **Geometric Sparse-Voxel Reconstruction**. Our method, abbreviated as GeoSVR, delivers high-quality surface reconstruction for intricate real-world scenes based on explicit sparse voxels. Our superiority is exhibited compared to the state-of-the-art approaches built upon Gaussian Splatting, which encounter rough, inaccurate, or incomplete recovery problems even with help from external estimators, excelling in delicate details capturing with high completeness and top-tier efficiency.

## Abstract

Reconstructing accurate surfaces with radiance fields has achieved remarkable progress in recent years. However, prevailing approaches, primarily based on Gaussian Splatting, are increasingly constrained by representational bottlenecks. In this paper, we introduce GeoSVR, an explicit voxel-based framework that explores and extends the under-investigated potential of sparse voxels for achieving accurate, detailed, and complete surface reconstruction. As strengths, sparse voxels support preserving the coverage completeness and geometric clarity, while corresponding challenges also arise from absent scene constraints and locality in surface refinement. To ensure correct scene convergence, we first propose a Voxel-Uncertainty Depth Constraint that maximizes the effect of monocular depth cues while presenting a voxel-oriented uncertainty to avoid quality degradation, enabling effective and robust scene constraints yet preserving highly accurate geometries. Subsequently, Sparse Voxel Surface Regularization is designed to enhance geometric consistency for tiny voxels and facilitate the voxel-based formation of sharp and accurate surfaces. Extensive experiments demonstrate our superior performance compared to existing methods across diverse challenging scenarios, excelling in geometric accuracy, detail preservation, and reconstruction completeness while maintaining high efficiency. Code is available at `https://github.com/Fictionarry/GeoSVR`.

39th Conference on Neural Information Processing Systems (NeurIPS 2025).

# 1 Introduction

Surface reconstruction from multi-view images has been a critical long-term problem in computer vision and graphics. In recent years, with the development of Neural Radiance Fields (NeRF) [43], impressive performances [71, 59, 38, 60] have been shown by combining volume rendering with signed distance functions (SDF) to learn implicit fields from input images, yet are mostly computationally expensive. More recently, with the rise of 3D Gaussian Splatting (3DGS) [32], surface reconstruction with explicit sparse representation is making rapid progress [25, 28, 78, 15, 11, 62], enabling efficient and high-quality geometry learning for a wider range of scenarios.

While significant advancements have been achieved in these 3DGS-based approaches, the methodological limitations are emerging as a bottleneck. One critical problem lies in the reliance on well-structured point clouds initialization. Typically provided by multi-view geometry (MVG) approaches [51, 52], the point clouds inevitably contain inaccurate and uncovered regions due to appearance ambiguities, which further aggravates difficulties for 3DGS to refine these challenging areas accurately in geometry, becoming an inherent flaw. This spatial incompleteness further hinders the full potential of rapidly evolving geometry foundation models [17, 69, 2] in attempts [15, 14, 36, 64], obstructing their ability to drive a quality revolution in surface reconstruction. Another key issue is the lack of clearly defined edges in the Gaussian primitives, making the geometry ambiguous from both the representation clarity [28, 55] and the calculation precision trade-offs [55, 48, 42].

Exploring another possibility, this paper presents **GeoSVR** that tames sparse voxels to achieve accurate and delicate surface reconstruction. Unlike previous explicit approaches based on 3DGS, we start with a recently proposed SVRaster [55] that combines sparse voxels with rasterization to efficiently refine the scene via level of details. Initialized with fully covered coarse voxels constantly, the full potential can be maintained to model any part inside the scene with completeness. And with clearly bounded voxels, geometric details can be better identified compared to the Gaussians or smooth neural fields. However, while obtaining distinct characteristics, challenges come correspondingly.

Despite competitive surface quality yielded by vanilla SVRaster, significant geometric distortion persists during the sparse voxels optimization, due to the absence of a strong structure prior like the point clouds used in 3DGS, hindering further surface refinement. To fully exploit the strength of the densely covered representation, we resort to the current rapidly evolving and increasingly well-established monocular depth as the geometry cue to provide dense and easy-to-fetch scene constraints. However, a key problem arises for our highly accuracy-required surface reconstruction task: how to effectively utilize this good but not perfect external constraint, while preserving well-reconstructed geometries from being hurt by errors to avoid quality degradation. To address this, we first adopt the patch-wise depth regularizer [35] to facilitate local geometry learning, and based on which, a Voxel-Uncertainty Depth Constraint is proposed, evaluating the geometric uncertainty of each voxel and adaptively determining the degree of reliance on external cues at pixel level. By modulating internal photometric and external depth supervisions for confident and ambiguous regions, our approach enables effective and robust scene constraints, even for well-reconstructed geometries.

Investigating the voxel-based surface formation, we then focus on geometric accuracy refinement and develop Sparse Voxel Surface Regularization. Since the gradients are shared only with the nearest neighbors, challenges exist in composing these extremely local and tiny sparse voxels to ideal surfaces. First, inspired by previous MVG-regularized approaches [20, 16, 11, 50, 12], we try to adopt the widely used explicit multi-view geometry constraint [26] to help build geometrically correct surfaces. Nevertheless, the sparse voxel's extreme locality made this plane-based geometry regularization less effective in enforcing a regional geometry constraint. To enlarge the refinement of per voxel, we conduct an interval sample to randomly drop out a portion of voxels to simplify the learned scene during geometry regularization, thus forcing each tiny voxel to keep a global geometry consistency. Second, from the perspective of voxel's surface representation, we introduce two voxel-wise regularizations: A Surface Rectification to restrict the surface formation to be aligned with a unique voxel to reduce depth bias; and according to the mentioned voxel uncertainty, a Scaling Penalty to eliminate the participation of the geometrically inaccurate large voxel in the surface formation. With the help of both, sharp and accurate surfaces are facilitated in the reconstruction.

In summary, our main contributions are as follows:

- An exploration GeoSVR to build explicit voxel-based framework for accurate surface reconstruction, taming sparse voxels to enable delicate and complete geometry learning.

- A Voxel-Uncertainty Depth Constraint that maximizes the utilization of external depth cue while avoiding quality degradation by the proposed voxel uncertainty evaluation, enabling effective and robust scene constraints for highly accuracy-required surface reconstruction.

- A Sparse Voxel Surface Regularization for surface geometric accuracy refinement, which enlarges the global geometry consistency constraint for tiny voxels and facilitates reconstructing sharp and accurate surfaces by regularizing the voxel-based surface formation.

- Extensive experiments on DTU, Tanks and Temples, and Mip-NeRF 360 datasets demonstrate that the proposed GeoSVR achieves superior performance compared to the state-of-the-arts in reconstructing accurate surfaces across diverse challenging scenarios, excelling in detail preservation and high completeness while maintaining computational efficiency.

## 2 Related Works

**Differentiable Radiance Fields.** In recent years, radiance fields have made significant progress in 3D reconstruction by learning scenes directly with differentiable rendering. Neural Radiance Fields (NeRF) [43] is one of the most important foundations, which uses a large MLP to memorize 3D scene and renders through differentiable volume rendering, yet is weak in efficiency. Later, hybrid representations come by proposing neural grids [56, 45, 27], plane decompositions [9, 10, 18, 8], or sparse voxels [19, 39, 74] with or without neural networks. However, a weakness is that these methods always assume all the grids are uniform in scale, limiting the quality and scalability. More recently, 3D Gaussian Splatting (3DGS) [32] represents radiance fields by a set of anisotropic 3D Gaussians and renders with differentiable splatting using rasterization, achieving remarkably successful balances between fast and high-quality scene reconstruction [40, 76, 49]. However, due to the complexity of tremendous intersected Gaussians, a view-inconsistent rendering problem is exhibited, and is hard to fix without large efficiency trade-offs [48, 42, 44]. Also, the reliance on sparse point clouds brings additional uncertainty. To this end, SVRaster [55] combines efficient rasterization with explicit non-uniform sparse voxels to achieve definite, robust, and high-quality scene representation, with less mandatory dependency on structure prior as well. Nevertheless, its potential for accurate geometry learning has not been fully explored, which is an open yet invaluable problem for 3D reconstruction.

**Surface Reconstruction with Learnable Fields.** Reconstructing surfaces from multi-view images has been a long-standing problem. While multi-view stereo-based methods [26, 81, 52, 70] rely on a modular pipeline with multiple decoupled stages, earlier neural approaches [72, 47] are proposed to represent surfaces implicitly with an MLP to learn geometry directly from images. Further advancements such as UNISURF [46], NeuS [59], VolSDF [71] represent implicit surfaces by signed distance functions integrated with differentiable volume rendering and achieve better reconstructed details. Based on these, methods with improvements like geometry regularizations [16, 77, 20, 58] and efficient grid representations [38, 65] extended the quality and available scenarios. However, the trade-off between training time and quality for complex scenes is still a serious challenge.

More recently, Gaussian-based surface reconstruction has arisen with 3DGS by offering better explicit geometry with much higher efficiency. SuGaR [25] first focuses on extracting Gaussians as mesh surfaces with alignment regularization. Then, more efforts appear by integrating 3DGS with SDF [75, 13, 41, 66, 79, 36] or improved representations [28, 78, 15], significantly progressing the surface quality. Specifically, 2DGS [28] and GSurfel [15] propose squeezing Gaussians as 2D surfels for a better aligned surface, and GOF creates an opacity field to allow dense and detailed mesh extraction. However, due to the loose geometry constraint and photometric ambiguities, challenges in accuracy still exist. For the SDF-integrated approaches, since additional MLP and also grid are usually required, problems of over-smoothness [79] and limitations for large unbounded scenes [41] may occur. To conquer the accurate reconstruction on challenging regions, following previous successes [20, 16, 12], PGSR inherits the idea of surfels and incorporates the multi-view geometry constraint [26], which is widely used in multi-view stereo, to regularize planar accuracy, but the production leans to be over-smooth, due to the unsharp Gaussians similarly in 2DGS [41, 75]. Meanwhile, some works [15, 14, 36, 62] attempt to resort to external geometry cues from geometry foundation models [17, 2, 69] for regularization. However, the performance is still far from what the cues could fully provide, mainly caused by the methodological bottleneck in the strict demand of high-quality initial points, for which the effect of special densification is also limited [11, 14]. Instead, this work explores another sparse voxel representation to escape the strict initialization requirement and pursue a clearer geometry representation, achieving superior accurate, detailed, and complete surface reconstruction.

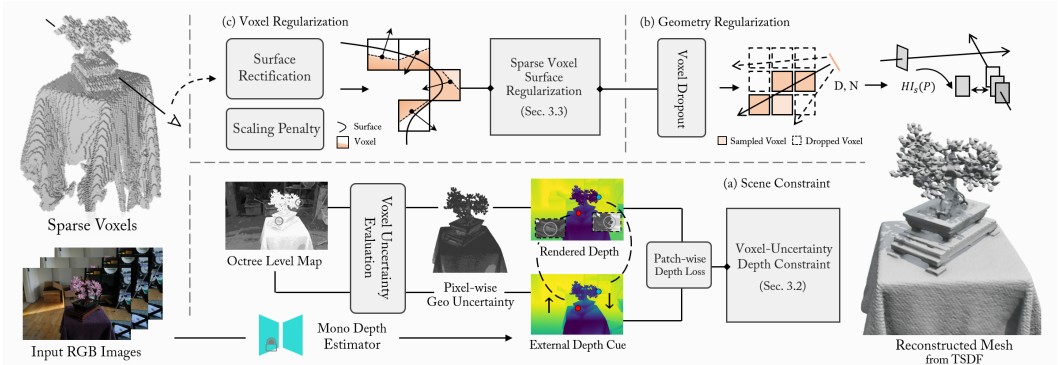

Figure 2: **Overview of GeoSVR**. Our method starts from constantly initialized sparse voxels, optimized with RGB images. (a) To enforce correct scene convergence while avoiding accuracy degradation, we apply Voxel-Uncertainty Depth Constraint by evaluating geometric uncertainty to determine the degree of reliance on monocular depth cue. (b) Voxel Dropout is introduced to enlarge the global geometry consistency for tiny voxels during the explicit geometry regularization. (c) For fine-grained surface refinement, we align the voxel-level density field to the surfaces with Voxel Regularization, facilitating accurate and sharp surface formation.

## 3 Method

### 3.1 Preliminaries: Sparse Voxels Rasterization

**Representation.** Sparse Voxels Rasterization (SVRaster) [55] represents scene with density field based on sparse voxels, which are organized in an Octree of the size $\mathbf{w}_s \in \mathbb{R}$ and center $\mathbf{w}_c \in \mathbb{R}^3$. Each voxel keeps a set of SH coefficients $\mathbf{v}_{sh}$ for voxel color, and densities $\mathbf{v}_{geo} \in [0, +\infty]^{2 \times 2 \times 2}$, separately on the eight voxel corners to model a trilinear inside density field for geometry. A voxel is identified with the index $v = \{i, j, k\}$ at Octree level $l$, and its size $\mathbf{v}_s$ and center $\mathbf{v}_c$ are given as:

$$\mathbf{v}_s = \mathbf{w}_s \times 2^{-l}, \quad \mathbf{v}_c = \mathbf{w}_c - 0.5 \times \mathbf{w}_s + \mathbf{v}_s \times v \qquad (1)$$

**Rendering.** During rendering, SVRaster adopts $\alpha$-blending similar to NeRF and 3DGS. Inside each voxel, SVRaster evenly samples $K$ points in the ray segment of length $\Delta t$ between the ray-voxel intersections, and composes voxel-wise $\alpha$ with trilinear interpolation $\mathrm{interp}(\cdot)$ by volume rendering. Then, the $\alpha$-blending is available to render the pixel-wise color $\mathbf{C}$ that corresponds to the ray:

$$\mathbf{C} = \sum_{i=1}^{N} T_i \alpha_i c_i, \ T_i = \prod_{j=1}^{i-1}(1 - \alpha_j); \quad \alpha = 1 - \exp(-\frac{\Delta t}{K} \sum_{k=1}^{K} \mathrm{interp}(\mathbf{v}_{geo}, \mathbf{q}_k)), \quad (2)$$

where $\alpha_i$ and $c_i$ are alpha and view-dependent color of the $i$-th intersected voxel, and $\mathbf{q}_k$ is the local position of the $k$-th sample point in the voxel. According to $\alpha$-blending, we can render the pixel-wise normal $\mathbf{N}$ and depth $\mathbf{D}$. The pixel-wise depth $\mathbf{D}$ can be given similarly via per-point distance rendering. For voxel's normal, the analytical gradient is calculated at the voxel center $\mathbf{q}_c$:

$$\mathbf{n} = \mathrm{normalize}\left(\nabla_{\mathbf{q}} \, \mathrm{interp}\left(\mathbf{v}_{geo}, \mathbf{q}_c\right)\right). \qquad (3)$$

**Adaptive Octree Control.** To adaptively adjust the scene Octree during training, SVRaster prunes the voxels with the least blending weight $T\alpha$, and accumulates an $\alpha$-weighted priority based on loss gradients to select the voxels that need to be subdivided to the next level to represent finer details.

**Challenges in Surface Reconstruction.** Despite strengths in geometric completeness and clarity, challenges exist correspondingly: **1)** With little native constraint, the optimization often encounters heavy geometry distortion and blocks further improvement. **2)** The impact scope of a single voxel could be quite local, which is unfavorable to accurate surface formation. Exploring tackling these two challenges, we present GeoSVR for high-quality voxel-based surface reconstruction, as in Figure 2

### 3.2 Voxel Geometric Uncertainty for Scene Constraint

Unlike previous approaches based on SDF [20, 77] or 3DGS [28, 78, 14, 11] that benefit from the structure constraints from geometric [1] or sparse points initialization [32], the highly expressive and constant-initialized sparse voxels require an essential scene constraint to effectively ensure the geometry converges to approximately correct surfaces, preparing for a further accuracy refinement.

**Problem in Monocular Depth Cue.** Inspired by previous works [77, 64], we turn attention to the increasingly well-established monocular depth [5, 21, 67, 69], which provides dense, efficient, and full-time available constraints for scene geometry optimization. Moreover, this dense cue natively matches the spatially complete voxels to fulfill its potential for compensating appearance ambiguities.

However, the problem of how to maximally utilize this attractive but not perfectly accurate prior in the highly accuracy-required surface reconstruction remains a long-standing difficulty. Despite considerable relevant studies [77, 58, 15, 14, 64, 62, 36], a solution is still absent to evaluate the learned geometry's confidence to determine external cue reliance, which causes only over-conservative strategies to be available, but could still degrade the quality by the included errors [77, 14].

**Voxel Geometric Uncertainty.** In this work, we aim to solve the problem by evaluating geometric uncertainty from the representational capability: **1)** In SVRaster, each voxel contains a trilinear density field to represent the geometry of the cube space with length of $\mathbf{v}_\mathrm{s} = \mathbf{w}_\mathrm{s} \times 2^{-l}$. Then, the accuracy for an under-captured geometry is strictly limited, negatively related to the level $l$ of corresponding voxels. **2)** During optimization, SVRaster progressively subdivides voxels at $l$ with largest gradients to the next level $l + 1$. Consequently, the voxels at lower levels denote either regions with fewer texture constraints or less view coverage, both associated with high uncertainties.

Inspired by these two tight couplings of uncertainty and voxel's level, we abstract a level-aware geometric uncertainty that explicitly correlates with Octree level $l$ to guide identifying scene constraint targets. For a voxel $v$ at level $l$, its base and geometric uncertainties $U_\mathrm{base}$ and $U_\mathrm{geom}$ are given by:

$$U_\mathrm{base}(l) = \frac{\mathbf{w}_\mathrm{s}}{\beta(l + l_0)}, \quad U_\mathrm{geom}(v) = U_\mathrm{base}(l) \cdot (1 - \exp(-\mathbf{v}_\mathrm{geo})), \tag{4}$$

where $\beta$ is a scaling factor, combined with Octree size $\mathbf{w}_\mathrm{s}$ as global scene scale. $l_0$ is the starting level. Geometric uncertainty $U_\mathrm{geom}(v)$ is composed of the level-dependent base uncertainty $U_\mathrm{base}$ and the voxel density, indicating a voxel at low level with critical geometry leads to higher uncertainty. Derived in the Appendix Section B, we simplify powers and exponents while preserving the same trend to prevent numerical blowup in later applying.

**Voxel-Uncertainty Depth Constraint.** Based on the uncertainty, we next design the constraint to enable effective and reliable monocular depth integration. To effectively apply the monocular depth as supervision, we resort to a patch-wise global-local depth loss [35] for better scale alignment and facilitating the geometry knowledge learning. Then, integrating the geometric uncertainty into the pixel-wise constraint, we first render an Octree level map $\mathbf{L}$ for efficient pixel-wise uncertainty calculation, directly gathering the volume density term of Eq. (4) with $\alpha$ of Eq. (2) via rasterization:

$$\mathbf{L} = \sum_{i=1}^{N} T_i \alpha_i l_i, \quad \alpha = 1 - \exp(-\frac{\Delta t}{K} \sum_{k=1}^{K} \mathrm{interp}(\mathbf{v}_\mathrm{geo}, \mathbf{q}_k)). \tag{5}$$

Next, converting uncertainty to weight, we produce a pixel-wise modulation on depth constraint. To ensure adaptive and robust constraints for various stages and scenarios, we obtain statistics of $\mathbf{L}$ to set the hyperparameters in Eq. (4). Specifically, for scale-independence, let the scale term of $\mathbf{w}_\mathrm{s}/\beta$ equal to per-view global level scale $\mathbf{w}_l = \max(\mathbf{L}) - \min(\mathbf{L})$, and set $l_0 = -\min(\mathbf{L})$ to define the coarsest level of the view. Then, derived from $U_\mathrm{geom}$, the geometry uncertainty weight $\mathbf{W}_\mathrm{unc}$ follows:

$$\mathbf{W}_\mathrm{unc} = \frac{\mathbf{w}_l}{\max(1, \mathbf{L} - \min(\mathbf{L}))}, \quad \mathbf{w}_l = \max(\mathbf{L}) - \min(\mathbf{L}) \tag{6}$$

Finally, given the estimated monocular depth $\widetilde{\mathbf{D}}$ as the constraint for the rendered depth $\mathbf{D}$, $\mathbf{W}_\mathrm{unc}$ is applied to the patch-wise depth loss $\mathcal{L}_\mathrm{D\text{-}patch}$ [35] for per-pixel constraint reweight:

$$\mathcal{L}_\mathrm{D\text{-}unc}(\mathbf{D}, \widetilde{\mathbf{D}}) = \mathbf{W}_\mathrm{unc} \cdot \mathcal{L}_\mathrm{D\text{-}patch}(\mathbf{D}, \widetilde{\mathbf{D}}). \tag{7}$$

As a result, Voxel-Uncertainty Depth Constraint $\mathcal{L}_\mathrm{D\text{-}unc}$ pays minimal attention to the voxels with low uncertainty to be confident of the native photometric constraint, while enhancing highly uncertain ones to rely on external cue for solving geometry ambiguities. The effect can be illustrated in Figures 2 and 6, where level map $\mathbf{L}$ is shown to obtain a more uniform range of values for better visual effect.

### 3.3 Sparse Voxel Surface Regularization

Despite the scene constraint exerted, a coarsely correct reconstruction does not exhibit the full potential of sparse voxels. Therefore, we next investigate the capability of sparse voxels for highly accurate surface formation under explicit geometry constraint and finer voxel-level regularizations.

**Geometry Regularization with Voxel Dropout.** Serving as an explicit and strict constraint, homography patch warping has shown great effect in classical MVS [22, 53, 81] and recent related works [20, 16, 11, 50, 12, 58], which we also try to apply in our method. Typically, considering a source view and a reference view with image $\mathbf{I}_s$ and $\mathbf{I}_r$, we warp the image point $\mathbf{x}'$ in the pixel patch $P$ of $\mathbf{I}_s$ to the image point $\mathbf{x}$ in $\mathbf{I}_r$ of the reference view by the plane-induced homography $\mathbf{H}$ [53]:

$$\mathbf{x} = \mathbf{H}\mathbf{x}', \ \ \mathbf{H} = \mathbf{K}_s(\mathbf{R}_s\mathbf{R}_r^T + \frac{\mathbf{R}_s(\mathbf{R}_s^T\mathbf{t}_s - \mathbf{R}_r^T\mathbf{t}_r)\mathbf{n}^T}{\mathbf{n}^T\mathbf{p}})\mathbf{K}_r^{-1}, \tag{8}$$

where $\mathbf{p}$ is the intersected 3D point calculated from depth $\mathbf{D}$, and the normal $\mathbf{n}$ is from Eq. 3. $\mathbf{K}$ is camera intrinsics, and $[\mathbf{R}, \mathbf{t}]$ is the extrinsics of each view. Then, an occlusion-aware NCC loss [11] is applied between the warped $P$ and its target in $\mathbf{I}_r$. However, we observe that despite improvements brought, this technique does not work as ideally as in previous approaches. Due to the extreme locality of the tiny voxels that connect to only the nearest neighbors by a few corners, the planar constraint becomes less effective, leading to redundant wrong structures being produced.

To solve this problem, our idea is to enlarge the regularization for each voxel by breaking these incorrectly organized geometries, enforcing the tiny voxels to obey a more global geometry consistency instead of only their own tiny scopes. During the process, we conduct an interval sample of the voxels with a random ratio in $[\gamma, 1]$ while calculating the full-scale depth $\mathbf{D}$ and normal $\mathbf{N}$. Therefore, only a subset of voxels is used to represent the scene, while the others are temporally dropped out. Then, the regularization enforces each voxel to respond to the geometry consistency of a larger area, including where the dropped-out voxels belong, for a forced break and correction of the ill geometries.

**Surface Rectification.** Subsequently, we focus on the bias between the trilinear voxel density field and the weight contribution in rendering, which causes misaligned surfaces from rendering and voxel density. As in Figure 3, due to the trilinear local-linked voxel fields, the density increase of one voxel will implicate the neighbors, resulting in decentralized densities that makes the highest rendering weight $w$ biased to the side regions but not the correct highest density position, like Figure 3 a.1.

(a) Surface Formation  (b) Surface Rectify

Figure 3: Illustration of Surface Rectification and the visualized process on voxels.

To this end, we propose Surface Rectification, a voxel-level regularization conducted during rendering. In the process, we first calculate an enter voxel alpha $\alpha_{p,e}$ and out $\alpha_{p,o}$ from the density of the intersected enter and out points $\mathbf{p}_e$, $\mathbf{p}_o$, denoted as red cross in Figure 3, to model the first-time intersection for surface checking, and select the voxels with critical change density between $\mathbf{p}_e$ and $\mathbf{p}_o$ cross a threshold $T_\alpha$ (set to 0.5 in this work) as the surface voxels $V_s$:

$$V_s = \{v \mid \alpha_{p,e} < T_\alpha < \alpha_{p,o}\}, \ \ \text{where} \ \ \alpha_{p,e/o} = 1 - \exp(-\Delta t \cdot \text{interp}(\mathbf{v}_{geo}, \mathbf{p}_{e/o})) \tag{9}$$

Then, for these voxels, we penalize the density at $\mathbf{p}_e$ but encourage at $\mathbf{p}_o$ to form a sharp segmentation of the surface and empty spaces, with a penalty term including the voxel's rendering contribution $w$:

$$\mathcal{R}_{rec} = w \cdot \mathbb{I}(v \in V_s) \cdot (\text{interp}(\mathbf{v}_{geo}, \mathbf{p}_e) - \text{interp}(\mathbf{v}_{geo}, \mathbf{p}_o)), \ \ \ w = T\alpha. \tag{10}$$

Since then, the surface in rendering can be rectified to be aligned to the density, as in Figure 3 a.2.

**Scaling Penalty.** Inspired by the voxel geometric uncertainty in Sec. 3.2, we present a simple yet effective regularizer that penalizes the voxels occupying a long sampling distance, which denotes a less accurate geometry modeling. Normalized with globally minimal voxel size $\min(\mathbf{v}_s)$, it follows:

$$\mathcal{R}_{sp} = w \cdot \text{interp}(\mathbf{v}_{geo}, \mathbf{q}_c) \cdot \max(0, \log_2(\frac{\Delta t}{\min(\mathbf{v}_s)})), \ \ \text{where} \ \ \mathbf{q}_c = (0.5, 0.5, 0.5). \tag{11}$$

### 3.4 Loss Function

The total objective is composed of the photometric loss $\mathcal{L}_{photo}$ from SVRaster, the depth constraint $\mathcal{L}_{D\text{-unc}}$ from Eq. (7), NCC loss for geometry regularization, and the voxel regularizations in Sec. 3.3:

$$\mathcal{L} = \mathcal{L}_{photo} + \eta\mathcal{L}_{D\text{-unc}} + \tau\mathcal{L}_{NCC} + \mu_1\mathcal{R}_{rec} + \mu_2\mathcal{R}_{sp}. \tag{12}$$

In this work, we set the weights of $\eta = 0.1$, $\tau = 0.01$, $\mu_1 = 10^{-5}$, and $\mu_2 = 10^{-6}$, respectively.

Table 1: **Quantitative Comparison on the DTU [31] Dataset**. Bests are in highlight. Our GeoSVR achieves the highest reconstruction quality on the Chamfer distance while retaining fast training.

| | | 24 | 37 | 40 | 55 | 63 | 65 | 69 | 83 | 97 | 105 | 106 | 110 | 114 | 118 | 122 | Mean | Time |
|---|---|---|---|---|---|---|---|---|---|---|---|---|---|---|---|---|---|---|
| Implicit | VolSDF [71] | 1.14 | 1.26 | 0.81 | 0.49 | 1.25 | 0.70 | 0.72 | 1.29 | 1.18 | 0.70 | 0.66 | 1.08 | 0.42 | 0.61 | 0.55 | 0.86 | > 12h |
| | NeuS [59] | 1.00 | 1.37 | 0.93 | 0.43 | 1.10 | 0.65 | 0.57 | 1.48 | 1.09 | 0.83 | 0.52 | 1.20 | 0.35 | 0.49 | 0.54 | 0.84 | > 12h |
| | Neuralangelo [38] | 0.37 | 0.72 | 0.35 | 0.35 | 0.87 | 0.54 | 0.53 | 1.29 | 0.97 | 0.73 | 0.47 | 0.74 | 0.32 | 0.41 | 0.43 | 0.61 | > 128h |
| | Geo-NeuS [20] | 0.38 | 0.54 | 0.34 | 0.36 | 0.80 | 0.45 | 0.41 | 1.03 | 0.84 | 0.55 | 0.46 | 0.47 | 0.29 | 0.36 | 0.35 | 0.51 | > 12h |
| | MonoSDF [77] | 0.66 | 0.88 | 0.43 | 0.40 | 0.87 | 0.78 | 0.81 | 1.23 | 1.18 | 0.66 | 0.66 | 0.96 | 0.41 | 0.57 | 0.51 | 0.73 | 6h |
| Explicit | 2DGS [28] | 0.48 | 0.91 | 0.39 | 0.39 | 1.01 | 0.83 | 0.81 | 1.36 | 1.27 | 0.76 | 0.70 | 1.40 | 0.40 | 0.76 | 0.52 | 0.80 | 0.2h |
| | GOF [78] | 0.50 | 0.82 | 0.37 | 0.37 | 1.12 | 0.74 | 0.73 | 1.18 | 1.29 | 0.68 | 0.77 | 0.90 | 0.42 | 0.66 | 0.49 | 0.74 | 1h |
| | SVRaster [55] | 0.61 | 0.74 | 0.41 | 0.36 | 0.93 | 0.75 | 0.94 | 1.33 | 1.40 | 0.61 | 0.63 | 1.19 | 0.43 | 0.57 | 0.44 | 0.76 | 0.1h |
| | GS2Mesh [62] | 0.59 | 0.79 | 0.70 | 0.38 | 0.78 | 1.00 | 0.69 | 1.25 | 0.96 | 0.59 | 0.50 | 0.68 | 0.37 | 0.50 | 0.46 | 0.68 | 0.3h |
| | VCR-GauS [14] | 0.55 | 0.91 | 0.40 | 0.43 | 0.97 | 0.95 | 0.84 | 1.39 | 1.30 | 0.90 | 0.76 | 0.92 | 0.44 | 0.75 | 0.54 | 0.80 | ~1h |
| | MonoGSDF [36] | 0.45 | 0.65 | 0.36 | 0.36 | 0.94 | 0.70 | 0.67 | 1.27 | 0.99 | 0.63 | 0.49 | 0.84 | 0.39 | 0.53 | 0.47 | 0.65 | hrs |
| | PGSR [11] | 0.36 | 0.57 | 0.38 | 0.33 | 0.78 | 0.58 | 0.50 | 1.08 | 0.63 | 0.59 | 0.46 | 0.54 | 0.30 | 0.38 | 0.34 | 0.52 | 0.5h |
| | **GeoSVR (Ours)** | 0.32 | 0.51 | 0.30 | 0.33 | 0.71 | 0.48 | 0.42 | 1.03 | 0.62 | 0.56 | 0.33 | 0.46 | 0.30 | 0.34 | 0.32 | 0.47 | 0.8h |

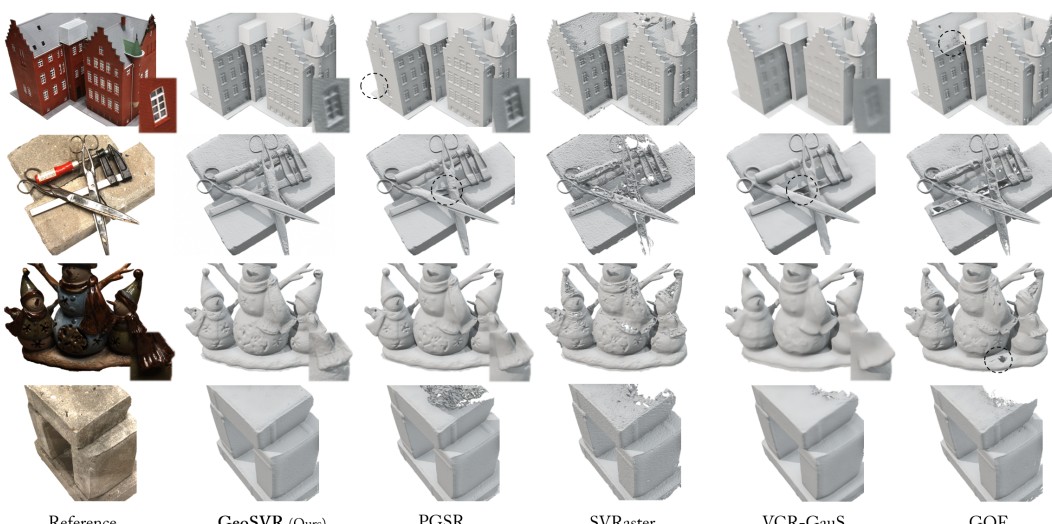

Reference     **GeoSVR** (Ours)     PGSR     SVRaster     VCR-GauS     GOF

Figure 4: **Reconstructed Mesh Visualization on the DTU [31] Dataset**. Our GeoSVR achieves superior reconstruction both in accuracy and completeness, handling difficult regions well by geometry cue constraints while still preserving fine-grained details. Better visualized with zoom in.

## 4 Experiments

**Implementation Details.** Our code is implemented with PyTorch and CUDA kernels, built upon SVRaster [55]. In the experiments, we train each model with 20,000 iterations, with the learning rates for density and SHs at degree 0 and the others of 0.05, 0.01, and 0.00025 in Adam [33] optimizer. We use DepthAnythingV2 [69] to provide the depth cues. The patch size of $7 \times 7$ is used for patch warping, and $\gamma$ in voxel dropout is set to 0.5 and 0.3 for DTU and TnT datasets. The Octree setups keep the same as in [55], and the prune interval is increase to 2,000 for finer expression. In our method, we use TSDF for mesh extraction. All experiments are conducted on RTX 3090 Ti GPUs.

### 4.1 Comparision

**Dataset.** We use the prevailing DTU, Tanks and Temples (TnT), and Mip-NeRF 360 datasets for evaluation. The scene selections of DTU and TnT are consistent with previous works [71, 59, 38, 28], preprocessed following 2DGS [28] and Neuralangelo [38]. The voxel size of TSDF is set to 0.002 for DTU and is calculated for TnT following PGSR [11]. The images in DTU and TnT are downsampled $2\times$, and in Mip-NeRF 360 are downsampled $2\times$ or $4\times$ following [32] for indoor and outdoor scenes.

**Baselines.** We take the state-of-the-art surface reconstruction approaches as baselines, including implicit (e.g., NeuS [59], Neuralangeo [38], Geo-NeuS [20]) and explicit methods (e.g., 2DGS [28], GOF [78], PGSR [11]). Among them, MonoSDF [77], GSurfel [15], VCR-GauS [14], GS2Mesh [62], and MonoGSDF [36] take external geometry cues from pre-trained depth and/or normal models for regularization. Basic representations like 3DGS [32] and SVRaster [55] are also included.

Table 2: **Quantitative Comparison on the Tanks and Temples** [34] **Dataset**. GeoSVR achieves the best on the F1 score, demonstrating superior reconstruction quality on various real-world scenarios.

| | Implicit | | | | Explicit | | | | | | |
| | NeuS | Neuralangelo | Geo-NeuS | MonoSDF | 2DGS | GOF | SVRaster | VCR-GauS | MonoGSDF | PGSR | **GeoSVR** |
|---|---|---|---|---|---|---|---|---|---|---|---|
| Barn | 0.29 | 0.70 | 0.33 | 0.49 | 0.41 | 0.51 | 0.35 | 0.62 | 0.56 | 0.66 | 0.68 |
| Caterpillar | 0.29 | 0.36 | 0.26 | 0.31 | 0.23 | 0.41 | 0.33 | 0.26 | 0.38 | 0.44 | 0.49 |
| Courthouse | 0.17 | 0.28 | 0.12 | 0.12 | 0.16 | 0.28 | 0.29 | 0.19 | 0.29 | 0.20 | 0.34 |
| Ignatius | 0.83 | 0.89 | 0.72 | 0.78 | 0.51 | 0.68 | 0.69 | 0.61 | 0.72 | 0.81 | 0.83 |
| Meetingroom | 0.24 | 0.32 | 0.20 | 0.23 | 0.17 | 0.28 | 0.19 | 0.19 | 0.25 | 0.33 | 0.37 |
| Truck | 0.45 | 0.48 | 0.45 | 0.42 | 0.45 | 0.59 | 0.54 | 0.52 | 0.62 | 0.66 | 0.66 |
| *Mean* | 0.38 | 0.50 | 0.35 | 0.39 | 0.30 | 0.46 | 0.40 | 0.40 | 0.47 | 0.52 | 0.56 |
| Time | >24h | >128h | >12h | 6h | 16m | 24m | 11m | 53m | 3h | 45m | 68m |

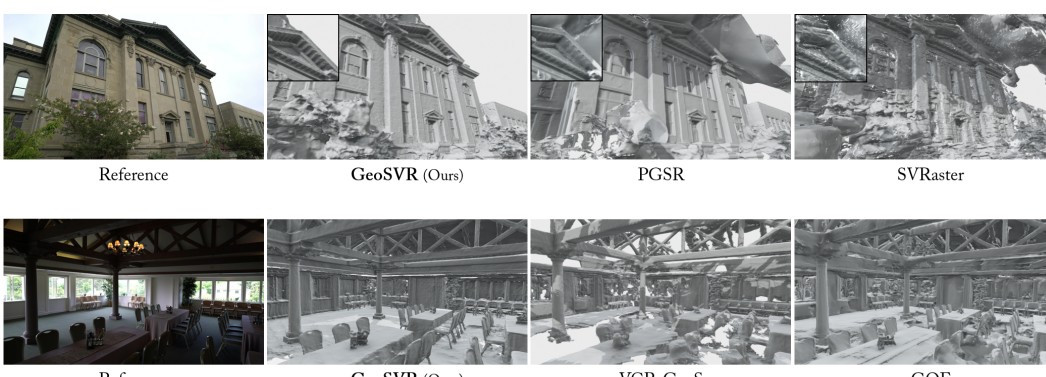

Figure 5: **Reconstructed Mesh Visualization on the Tanks and Temples** [34] **Dataset**. Our GeoSVR stands out by reconstructing accurate surfaces even for difficult scenes like complex buildings and weak texture regions, delivering intricate details as well as precise flats.

**Surface Reconstruction.** To evaluate surface reconstruction performance, we make comparisons on the DTU and TnT datasets with Chamfer distance and F1-score. The results are reported in Table 1 and 2. On the DTU dataset, our method outperforms all the baselines in the overall accuracy, exceeding previous SDF and 3DGS-based SOTA methods Geo-NeuS and PGSR, and also all the methods leveraging external geometry cues as well. On the TnT, our method also achieves the best in F1-score, and gets better results in most scenes compared to the SDF-based Neuralangelo, monocular depth-hinted (i.e., DepthAnythingV2) MonoGSDF, and geometry regularized PGSR.

On the two datasets, our method also retains fast training comparable to the 3DGS-based methods. In Figure 4 and 5, we visualize the reconstructed meshes from ours and competitive baselines. Our productions obtain both the best accuracy and completeness. And due to the basis of the initial prior-free and densely covered sparse voxels, GeoSVR can handle the reflective regions and areas with insufficient coverage, where the 3DGS-based methods are limited, due to insufficient initialization points. Additionally, GeoSVR performs better than previous geometry cue-reliant methods (e.g., VCR-GauS) that may lead to oversmoothing and underfitting.

Table 3: Quantitative Results on Mip-NeRF 360 Dataset. The best scores for surface reconstruction methods are highlighted with colors.

| | | Outdoor Scene | | | Indoor Scene | | |
| | | PSNR ↑ | SSIM ↑ | LPIPS ↓ | PSNR ↑ | SSIM ↑ | LPIPS ↓ |
|---|---|---|---|---|---|---|---|
| NVS | NeRF | 21.46 | 0.458 | 0.515 | 26.84 | 0.790 | 0.370 |
| | Deep Blending | 21.54 | 0.524 | 0.364 | 26.40 | 0.844 | 0.261 |
| | Instant NGP | 22.90 | 0.566 | 0.371 | 29.15 | 0.880 | 0.216 |
| | Mip-NeRF 360 | 24.47 | 0.691 | 0.283 | 31.72 | 0.917 | 0.180 |
| | 3DGS | 24.67 | 0.728 | 0.240 | 30.96 | 0.924 | 0.187 |
| | SVRaster | 24.68 | 0.738 | 0.206 | 30.65 | 0.927 | 0.161 |
| Surface Recon. | BakedSDF | 22.47 | 0.585 | 0.349 | 27.06 | 0.836 | 0.258 |
| | SuGaR | 22.93 | 0.629 | 0.356 | 29.43 | 0.906 | 0.225 |
| | 2DGS | 24.34 | 0.717 | 0.246 | 30.40 | 0.916 | 0.195 |
| | GOF | 24.82 | 0.750 | 0.202 | 30.79 | 0.924 | 0.184 |
| | VCR-GauS | 24.31 | 0.707 | 0.280 | 30.53 | 0.921 | 0.184 |
| | PGSR | 24.76 | 0.752 | 0.203 | 30.36 | 0.934 | 0.147 |
| | GeoSVR (Ours) | 24.83 | 0.738 | 0.218 | 30.46 | 0.921 | 0.172 |

**Appearance Reconstruction.** Achieving accurate geometry reconstruction, our method maintains the capability for high-quality novel view synthesis as well. In Table 3, we compare our methods on the Mip-NeRF 360 dataset with the baselines in the aspect of rendering quality. Our method exhibits competitive performance among the surface reconstruction methods as well as the NVS-specific baselines such as our basis SVRaster. Due to the lack of geometry ground-truth, we do not evaluate the surface reconstruction quality. Qualitative comparisons can be found in the Appendix.

## 4.2 Ablation Study

In this section, we verify the effect of our designs on the Tanks and Temples [34] dataset and report the mesh reconstruction metrics. The quantitative scores are reported in Table 4. As references, the reproduced SVRaster and PGSR with TSDF are reported in the comparison. Additionally, we summarize the baselines with external cues in Table 5 to exhibit the effect of the methodology itself.

Table 4: **Ablation Study** on the TnT Dataset.

| Items | Settings | Precision ↑ | Recall ↑ | F1-Score ↑ |
|---|---|---|---|---|
| **A.** | SVRaster (Base) | 0.383 | 0.421 | 0.397 |
| | PGSR (Reference) | 0.509 | 0.560 | 0.527 |
| | PGSR + Patch-wise Depth (Reference) | 0.517 | 0.576 | 0.538 |
| **B.** | A. + Sparse Points | 0.363 | 0.409 | 0.382 |
| | A. + Inverse Depth | 0.383 | 0.421 | 0.398 |
| | A. + Patch-wise Depth | 0.474 | 0.438 | 0.449 |
| **C.** | B. + Multi-view Reg. | 0.520 | 0.568 | 0.538 |
| | B. + Multi-view Reg. + Voxel Dropout | 0.533 | 0.569 | 0.546 |
| **D.** | C. + Surface Rectif. | 0.536 | 0.572 | 0.549 |
| | C. + Surface Rectif. + Scaling Penalty | 0.538 | 0.577 | 0.552 |
| **E.** | D. + Voxel-Uncertainty Depth (Ours) | 0.549 | 0.581 | 0.560 |

Table 5: **Accuracy Comparison** to Baselines with External Cues.

| Method | Geo. | Init.& Cues | TnT F1.↑ | DTU Cf.↓ |
|---|---|---|---|---|
| MonoSDF [77] | SDF | Mono Depth & Normal | 0.39 | 0.73 |
| GSurfel [15] | GS | SfM pts Mono Normal | - | 0.88 |
| GS2Mesh [62] | GS | SfM pts Stereo Depth | - | 0.68 |
| VCR-GauS [14] | GS | SfM pts Mono Normal | 0.40 | 0.80 |
| MonoGSDF [36] | GS+ SDF | SfM pts Mono Depth | 0.47 | 0.65 |
| **Ours** | Voxel | Mono Depth | **0.56** | **0.47** |

**Scene Constraint.** Scene constraint dominates an essential start for further refinement. In Table 4, we observe that regularizations of sparse depth from SfM points and monocular depth with inverse loss both help less, while the patch-wise depth loss of Table 4 B breaks through to improve the geometry effectively. A step further, even though the reconstruction already achieves a high quality (0.552 in F1), our *Voxel-Uncertainty Depth Constraint* still remarkably recognizes the uncertain regions and refines the geometry and preserving the well-reconstructed parts, as shown in Figure 6 and Table 4 E.

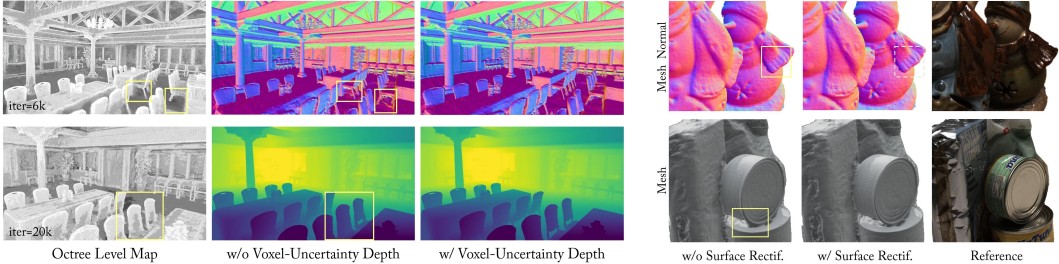

Figure 6: **Qualitative Studies** for the Voxel-Uncertainty Depth. Recognizing regions with uncertain voxel geometry, the challenging inaccurate surfaces can be effectively fixed.

Figure 7: **Qualitative Studies** for the Surface Rectification. Facilitation for sharp and accurate surfaces is made.

**Multi-view Regularization.** Based on the scene constraint, we then analyse the multi-view regularization part. Consistent with the conclusions like in previous works [20, 16, 11], adding the explicit multi-view geometry objective can hugely improve the geometry accuracy, yet by relieving the local trap of sparse voxels, our *Voxel Dropout* strategy further improves the multi-view consistency to a higher level and exceeds the patch-warping regularized reference method with monocular depth.

**Voxel Regularization.** To further facilitate the surface-depth consistency, we apply the *Surface Rectification* and *Scaling Penalty* for the voxels to get finer surfaces. As shown in Table 4 D and Figure 7, the voxel regularization designs benefit the formation of accurate surfaces from the perspective of voxel-based representation, therefore improving the geometry both quantitatively and qualitatively.

## 5 Conclusion

In this work, we have presented GeoSVR, an explicit voxel-based framework that explores and extends the under-investigated potential of sparse voxels to deliver accurate, detailed, and complete surface reconstruction with high efficiency. Our study first analyzes voxel uncertainty in geometry representation to distinguish the confidence of learned geometry, enabling effective and robust scene constraint from external cues. Next, we investigate the problem of voxel-based surface refinement, reconstructing surfaces with superior quality by our solution. In the future, it will be interesting to explore enhancing voxel's globality to conquer challenges like varying lights and textureless regions.

## Acknowledgments and Disclosure of Funding

This work is supported by the National Natural Science Foundation of China 62276016, 62372029, and Key R&D Program of Jiangxi Province, China (20232BBE50019). Lin Gu is supported by JST Moonshot R&D Grant Number JPMJMS2011 Japan.

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

# A Ablation Study

## A.1 Scene Constraint

To better verify and demonstrate the effect of scene constraint, we conduct an additional ablation study by solely disabling the scene constraint on our full GeoSVR and ablate its effect with more monocular models, including DepthAnything [68], Depth-Pro [6], and DepthAnythingV2 [69].

As analysed in our main paper, unlike 3DGS or SDF that natively obtain a geometric hypothesis for better convergence, the absence of scene constraint of SVRaster leads to undesirable and heavily distorted surfaces.

Table 6: **Additional Ablation Study on Scene Constraint.** The absence of scene constraint leads to obvious distorted geometry (red). This drawback can be solved via our proposal.

| Backbone | Monocular Model | Uncertainty | Precision ↑ | Recall ↑ | F1-Score ↑ |
|---|---|---|---|---|---|
| PGSR | None | N/A | 0.509 | 0.560 | 0.527 |
| | DepthAnythingV2 | N/A | 0.517 | 0.576 | 0.538 |
| GeoSVR | None | N/A | 0.511 | 0.549 | 0.523 |
| | DepthAnything | ✗ | 0.526 | 0.568 | 0.540 |
| | | ✓ | 0.539 | 0.574 | 0.551 |
| | DepthPro | ✗ | 0.537 | 0.574 | 0.549 |
| | | ✓ | 0.546 | 0.579 | 0.557 |
| | DepthAnythingV2 | ✗ | 0.538 | 0.577 | 0.552 |
| | | ✓ | 0.549 | 0.581 | 0.560 |

Although a competitive accuracy on partial regions can be gained after applying our efforts of Sparse Voxel Surface Regularization, the distorted geometry leads to a lot of inaccurate reconstructions and drags the overall performance improvement compared to the SOTA 3DGS-based approach PGSR [11], as quantitatively and qualitatively shown in Table 6 and Figure 8. By introducing monocular depth as a solution, it can be observed that this drawback of the representation has been well addressed.

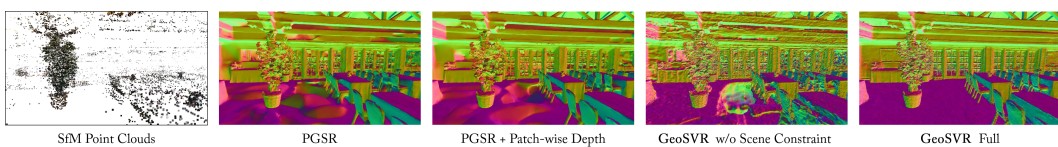

SfM Point Clouds     PGSR     PGSR + Patch-wise Depth     GeoSVR w/o Scene Constraint     GeoSVR Full

Figure 8: Comparison of Challenging Region Reconstruction without/with Scene Constraint.

Despite the rich geometric cues provided, previous 3DGS-based approaches are much more limited in exploiting monocular depths for improvements, which demonstrates the advantage of our explored sparse voxels. For demonstration, we apply the monocular depth with our used patch-wise loss to the SOTA PGSR and adjust the coefficient to fit the best F1-score, and visualize the reconstructed surfaces in Figure 8. It can be observed that for challenging areas that are difficult to reconstruct through multi-view consistency, 3DGS-based approaches can hardly recover the accurate geometry even when applying monocular depth constraints, mainly limited by their heavy reliance on the initial SfM points. Notably, PGSR also applies a specific densification technique AbsGS [73] to relieve this limitation, but this is still marginal for such cases. Instead, when just applying the less advanced DepthAnything, our method gets much larger improvements, especially equipped with the proposed Voxel-Uncertainty Depth Constraint. With DepthPro that is less robust for outdoor scenes, our method can still retain high-quality reconstruction, exhibiting the effectiveness and robustness of our method.

## A.2 Voxel Dropout

In the main paper, we have quantitatively verified the effect of Voxel Drouput, which improves the release of the power of geometry regularization for the local voxels significantly. Due to the space limitation of the main paper, we supplement the qualitative comparisons here to help demonstrate its effect in Figure 9. As shown, different from the situations in SDF or 3DGS where smooth surfaces are reconstructed by applying the homography patch warping, several redundant geometric structures remain in our local voxel-based representation when not applying

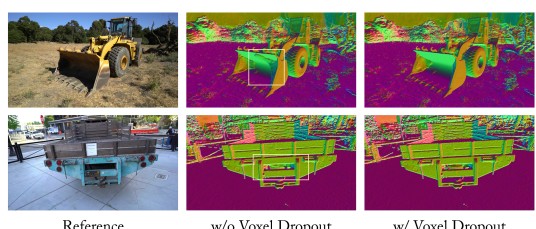

Reference     w/o Voxel Dropout     w/ Voxel Dropout

Figure 9: **Effect of Voxel Dropout Strategy**.

voxel dropout. Analysing these geometric artifacts, we found that despite the geometry regularization can recognize and penalize these regions, the effect is marginal because the effect scope is quite small for a tiny voxel and it does not receive the gradient from the distant neighbors, whereas it can not move as the primitives in 3DGS, making local minima between the appearance supervision and geometry regularization. Getting inspiration from the classic dropout [54] that solves the overfitting from complex parameters, which is similar to our problem, we simplify the voxel-based scene representation by dropping out parts of the voxels, and make each voxel obtain a large responsibility for geometric consistency during geometry regularization. This proposal finally relieves the problem.

### A.3 Surface Rectification

In Table 4, we ablate the Surface Rectification on the TnT datasets to show its effect, and report the comparison on DTU in Figure 7. Acting as a fine-grained regularization technique on tiny voxels with critical density changes, its effect can be better exhibited on the DTU dataset, which focuses on the evaluation for highly accurate surface reconstruction. For better

Table 7: Additional Ablation Study on Surface Rectification on the DTU [31] dataset.

| Setting | d2s $\downarrow$ | s2d $\downarrow$ | cf-dist $\downarrow$ |
|---|---|---|---|
| w/o Surface Rectif. | 0.433 | 0.521 | 0.477 |
| w/ Surface Rectif. | 0.426 | 0.511 | 0.468 |

demonstration, we report the quantitative ablation results on DTU in Table 7. Compared to the TnT dataset, the improvement on it is more significant to bring about a 0.01 improvement on Chamfer distance. This is even close to the entire gap between some previous methods (e.g., 0.51 from Geo-Neus v.s. 0.52 from PGSR) in Table 1, especially considering the quality is already quite close to the ground truth, which demonstrates our technique's effectiveness in accurate surface refinement.

## B  Derivation of Voxel Geometric Uncertainty

Here we provide the detailed derivation of the Voxel Geometric Uncertainty in Eq. (4).

**Derivation**. Review the representation, the scene geometry is represented with the composition of non-overlapping trilinear voxels, of which the density inside a voxel is trilinearly weighted by the 8 corner points with a density value of each. Therefore, the geometry representation capability for a single voxel is constant and scale-independent, which we denote as a constant value $G_{\mathrm{max}}$ to indicate the maximum quantity of information of each voxel for geometry representation.

Then, consider a local region of the quantity of information $\rho_{\mathrm{geo}}$ in geometry per unit volume that needs to be learned. For a voxel with the length of $\mathbf{v}_{\mathrm{s}}$ in it, the desired quantity of information $Q$ for representation should be:

$$Q = \rho_{\mathrm{geo}} \cdot \mathbf{v}_{\mathrm{s}}^3, \quad s.t. \ \mathbf{v}_{\mathrm{s}} = \mathbf{w}_{\mathrm{s}} \times 2^{-l} \quad \Rightarrow \quad Q = \rho_{\mathrm{geo}} \cdot \left(\frac{\mathbf{w}_{\mathrm{s}}}{2^l}\right)^3. \tag{13}$$

In most situations, the sparse voxel model can not fully capture the geometry information with unlimited resolutions due to the limited computational resources. Therefore, for the underfitted regions, the maximum information retention ratio $\eta_{\mathrm{max}}$ can be given:

$$\eta_{\mathrm{max}} = \frac{G_{\mathrm{max}}}{Q} = \frac{G_{\mathrm{max}}}{\rho_{\mathrm{geo}}} \cdot \left(\frac{2^l}{\mathbf{w}_{\mathrm{s}}}\right)^3 \in (0, 1). \tag{14}$$

After getting the ideal upper-bound, we focus on the common cases for the uncertainty.

First, considering Eq. 14 only gives the maximum retention ratio, while we try to estimate the actual voxel uncertainty, the local target for different voxels should be considered. Then, based on the retention ratio $\eta_{\mathrm{max}}$, our goal is to model a proper base to weight the uncertainty that is only relevant to the voxel's characteristic. Therefore, we build the voxel uncertainty based on an inverse $\eta$ to indicate the information loss, which ensures the weight does not shrink to 0 to cause failure when the ideal $G_{\mathrm{max}}$ meets the target $Q$, and introduce a coefficient $g(v)$ to adjust the local target for different voxels. The voxel geometric uncertainty $U$ in the initial follows:

$$\eta'_{\mathrm{max}} = \frac{G_{\mathrm{max}}}{Q \cdot g(v)} = \frac{G_{\mathrm{max}}}{\rho_{\mathrm{geo}} \cdot g(v)} \cdot \left(\frac{2^l}{\mathbf{w}_{\mathrm{s}}}\right)^3, \quad U(v) = \eta'^{-1}_{\mathrm{max}} = \frac{1}{G_{\mathrm{max}}} \cdot \left(\frac{\mathbf{w}_{\mathrm{s}}}{2^l}\right)^3 \cdot \rho_{\mathrm{geo}} \cdot g(v). \tag{15}$$

**Coefficient Definition and Approximation.** Turn to the real situations, since it's difficult to precisely define the value of constants $G_{\mathrm{max}}$ and $\rho_{\mathrm{geo}}$, and variance $g_v$, we denote the constant $G_{\mathrm{max}}$ as a

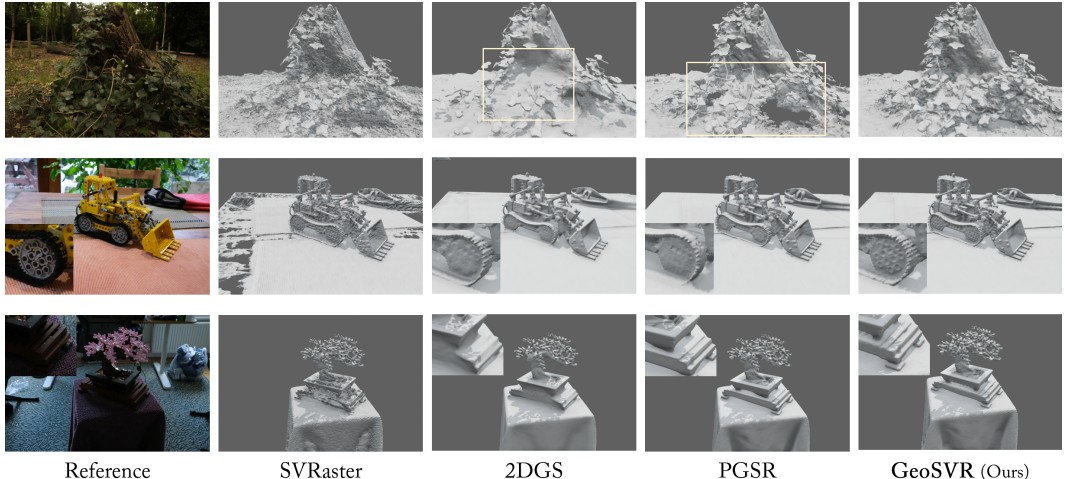

| Reference | SVRaster | 2DGS | PGSR | GeoSVR (Ours) |

Figure 10: **Qualitative Comparison on Mip-NeRF 360** [3] **dataset**. Our GeoSVR provides more detailed and complete surface reconstruction for the real-world captured scenes.

coefficient $\beta$ that relates to a glocal geometry scale, and take the sampled voxel density to approximate represent the term $\rho_{\text{geo}} \cdot g(v)$ to reflect the local quantity of geometric information, reasonably assuming the already learned densities in voxels have already been approximately accurate during the past optimization and are positively correlated to their local quantity of geometric information. Therefore, based on Eq. (15), the approximated voxel geometric uncertainty $U_{\text{appr}}$ is given by:

$$U_{\text{appr}}(v) = \frac{1}{\beta} \cdot \left( \frac{\mathbf{w}_{\text{s}}}{2^l} \right)^3 \cdot \left( 1 - \exp\left( -\mathbf{v}_{\text{geo}} \right) \right), \tag{16}$$

where $\text{interp}(\cdot)$ is not shown to indicate non-specific sampling. Considering further probable involvement in various calculations, given the values of Octree level $l$ may increase to be up to 16 or even larger, $U_{\text{appr}}$ can easily cause numerical blowup to an extremely extensive data range due to its contained power and exponent in $(2^l)^3$, and thus brings instability. Therefore, keeping the same trend and also a suitable data range as a weight for later, we cancel the overall power of $3$ and replace the term $1/2^l$ with $1/l$. To additionally compensate for the degree of value variation, we introduce a bias $l_0$ on the Octree level to help control the shape of the function. Consequently, the final formulations of the Voxel Geometric Uncertainty in Eq. (4) are derived:

$$U_{\text{base}}(l) = \frac{\mathbf{w}_{\text{s}}}{\beta(l + l_0)}, \quad U_{\text{geom}}(v) = U_{\text{base}}(l) \cdot \left( 1 - \exp\left( -\mathbf{v}_{\text{geo}} \right) \right). \tag{17}$$

## C Qualitative Comparison on Mip-NeRF 360

Here we supplement the qualitative comparison on Mip-NeRF 360 [3] dataset. As shown in Figure 10, our GeoSVR provides more detailed and complete surface reconstruction for the real-world captured images. As discussed in the main paper, both 2DGS and PGSR suffer from the geometry representation ambiguity from the Gaussians, and thus lead to over-smooth surfaces and a lack of details. Instead, our method overcomes the geometry distortion in SVRaster while preserving high-quality details with completeness to deliver accurate surface reconstruction.

## D Experimental Details

### D.1 Datasets

We use the DTU [1], Tanks and Temples (TnT) [2], and Mip-NeRF 360 [3] datasets for evaluation. Specifically, we follow the previous works to select 15 scans with ids of 24, 37, 40, 55, 63, 65, 69,

---

[1] https://roboimagedata.compute.dtu.dk/?page_id=36
[2] https://www.tanksandtemples.org/
[3] https://jonbarron.info/mipnerf360/

83, 97, 105, 106, 110, 114, 118, 122, and use the half-resolution images as the training data. The DTU dataset used in the experiment is preprocessed from 2DGS [28] through COMLAP [52, 51] [4]. In TnT dataset, we follow previous work to use 6 high-quality scenes from the Training Data split that provides publicly accessible ground truth for evaluation. The camera poses and scene boundary are translated by the script provided by Neuralangelo [38] [5]. For Mip-NeRF 360, we use all 9 scenes for evaluation. The images are downsampled 2× or 4× following [32] for indoor scenes ("bonsai", "counter", "kitchen", "room") and outdoor scenes ("bicycle", "garden", "flowers", "stump", "treehill"). The camera poses are provided along with the dataset.

### D.2   Baselines

In experiments, our baselines mainly involve: 1) implicit SDF-based methods: NeuS [59], Neuralangeo [38], Geo-NeuS [20], MonoSDF [77], and 2) explicit methods: 2DGS [28], GOF [78], GS2Mesh [62], VCR-GauS [14], PGSR [11]), MonoGSDF [36] and SVRaster [55]. In the latter, SVRaster is based on sparse voxels and others are based on 3DGS, while MonoGSDF also uses a hybrid SDF.

**Baselines Involving Geometry Models.** Among them, MonoSDF [77], GSurfel [15], VCR-GauS [14], GS2Mesh [62], and MonoGSDF (named G2SDF in the earlier [6]) [36] take external geometry cues from pre-trained depth and/or normal models for regularization. Specifically, MonoSDF uses pre-trained Omnidata [17] models to estimate monocular depth and normal, and GSurfel uses normal from Omnidata for supervision. GS2Mesh takes a pre-trained stereo model [80] to extract surfaces from a well-trained 3DGS model directly from the synthesized views. VCR-GauS leverages monocular normal [2] and conducts a multi-view check to estimate the estimation confidence of the normal maps for better regularization. MonoGSDF takes monocular depth from DepthAnythingV2 [69], which is the same as we use, to supervise the rendered depth with some learnable adjustment terms.

**Source of Results.** For qualitative results, we report the official scores from the published or up-to-date arXiv papers if available. For Geo-NeuS that does not have an official score on TnT dataset, we take the results reported by Neuralangelo for comparison. For qualitative results, we prioritize using the officially provided checkpoints or results if available. Otherwise, we use the official codebase to reproduce the results following the corresponding scripts on the same processed datasets as used for our method, which does not contain error poses like the processed TnT from 2DGS and GOF, for fairness. The reported training time is from the corresponding papers. And since MonoGSDF has not reported the training time on DTU nor open-sourced their code, we marked it as "hrs" through an approximated estimation from the provided training time of TnT.

### D.3   Metrics

Following prevailing settings [71, 59, 38, 28, 78], we use Champer distance to measure the accuracy for DTU dataset, and F1-Score for the overall quality for TnT. Especially, we take the off-the-shelf evaluation toolkits [7] [8] with the corresponding version of dependencies (e.g., Open3D 0.9.0 for TnT) in measurement for fairness. For Mip-NeRF 360 dataset, we inherit the metrics with implementations used in 3DGS [32] to keep aligned with previous works [78, 28, 14, 11].

## E   Inference Speed

Besides delivering high-quality surface reconstruction, GeoSVR also retains high efficiency. While the training times are reported in the main paper, we list the rendering speed on different datasets in the experiments in Table 8. It's shown that our method achieves a fast inference speed,

Table 8: Average Rendering Speed.

| Dataset | 360 [3] | DTU [31] | TnT [34] |
|---------|---------|----------|----------|
| FPS | 83.1 | 143.8 | 142.0 |

inheriting from SVRaster [55] by keeping its concise representation without introducing any heavy modules at inference. The typical time of mesh extraction is within minutes, depending on the required scale. Overall, GeoSVR achieves a top-tier efficiency competitive with GS-based methods.

---

[4] https://huggingface.co/datasets/dylanebert/2DGS
[5] https://github.com/NVlabs/neuralangelo/blob/main/DATA_PROCESSING.md
[6] https://arxiv.org/abs/2411.16898v1
[7] https://github.com/isl-org/TanksAndTemples/tree/master/python_toolbox/evaluation
[8] https://github.com/hbb1/2d-gaussian-splatting/tree/main/scripts/eval_dtu

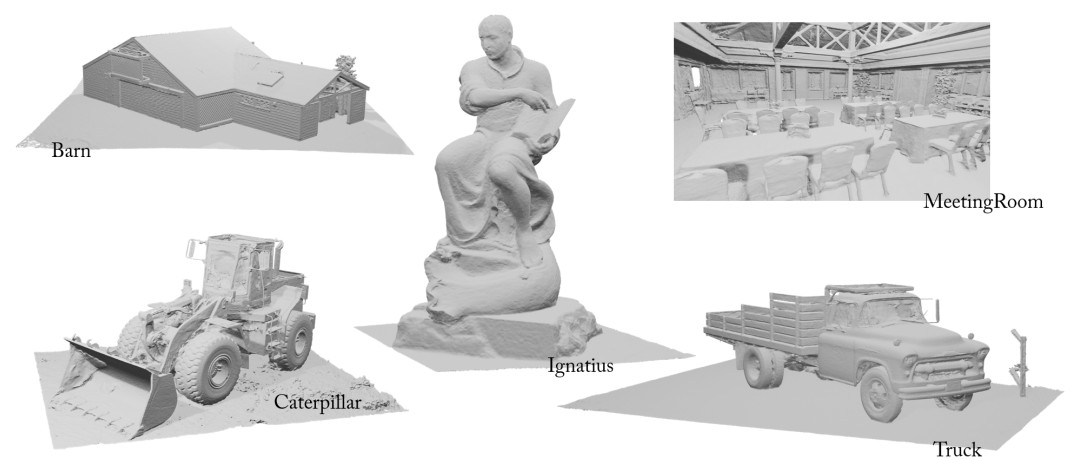

Figure 11: **Visualization of Reconstructed Meshes on TnT** [34] **Dataset.**

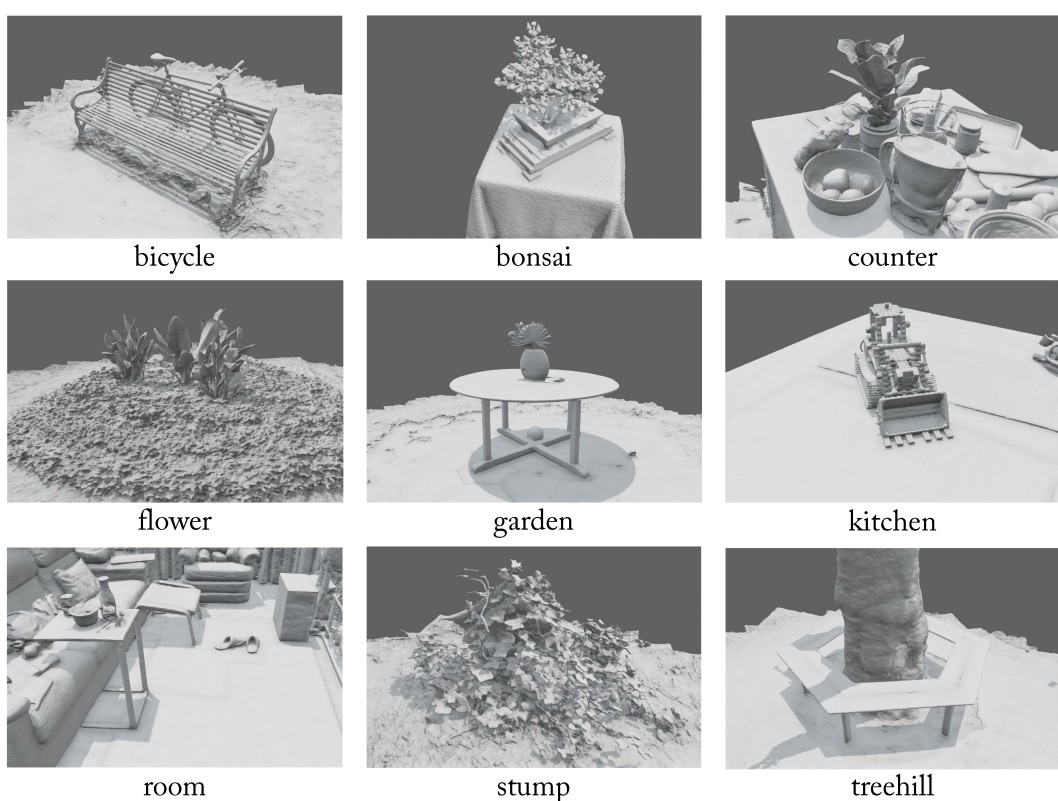

Figure 12: **Visualization of Reconstructed Meshes on Mip-NeRF 360** [3] **Dataset.**

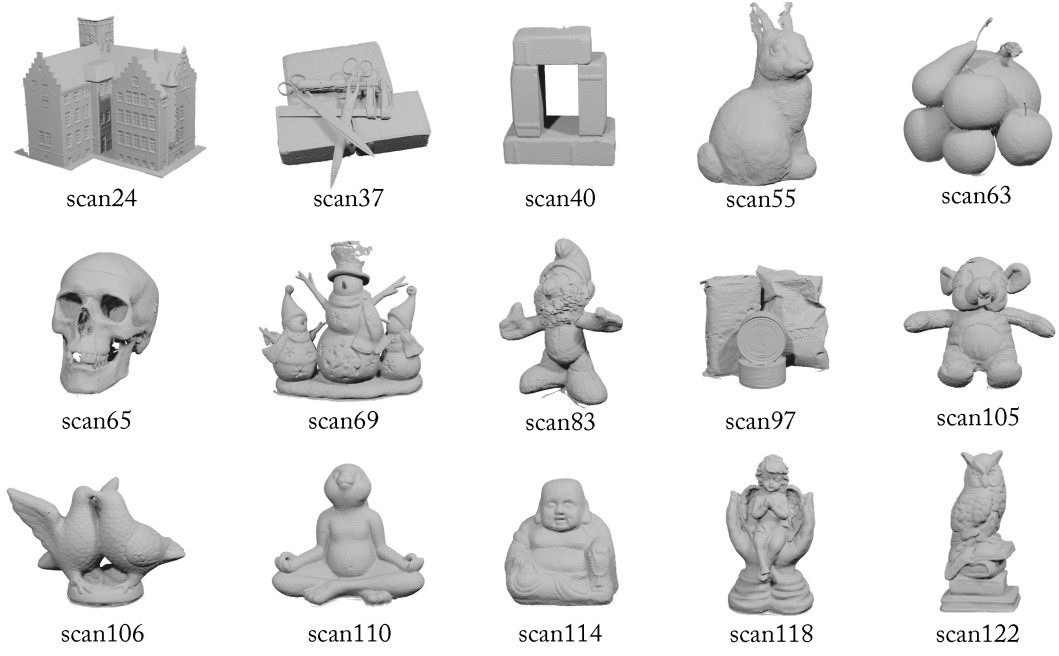

Figure 13: **Visualization of Reconstructed Meshes on DTU** [31] **Dataset.**

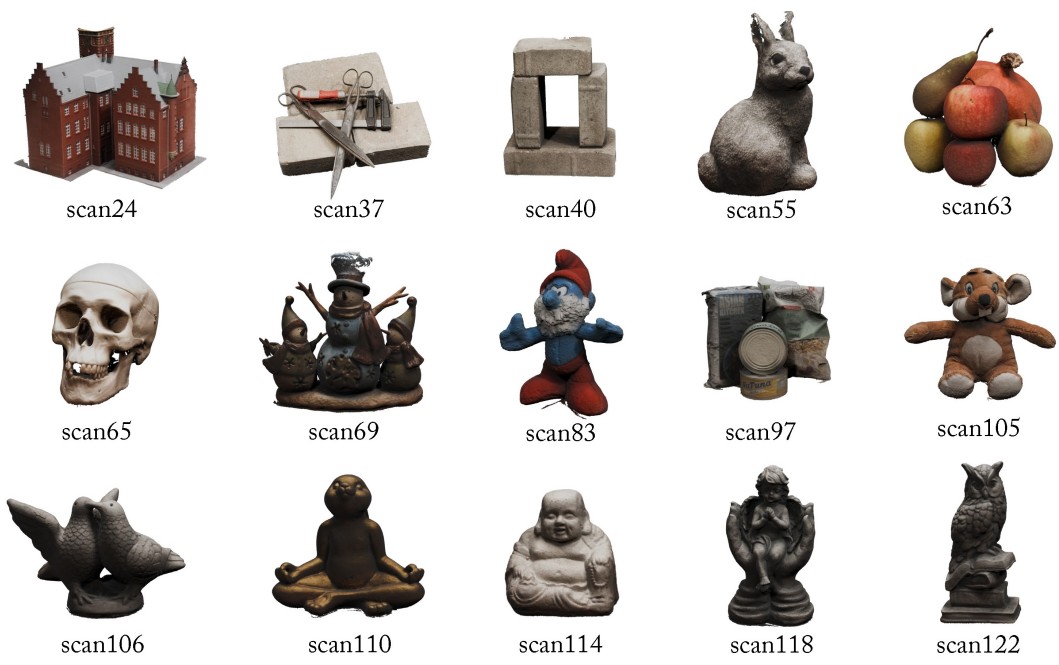

Figure 14: **Visualization of Reconstructed Meshes with Vertex Color on DTU** [31] **Dataset.**

## F Efficiency Analysis

In Table 9, we report the efficiency metrics corresponding to the ablation study Table 4. From the results, it can be observed that all of the components maintain high efficiency in all aspects, including inference FPS, memory, and the required number of voxels, except the multi-view regularization that contributes most to the increasing training time consumption. According to our analysis, this is mainly caused by the less efficient code implementation, which we plan to solve in the future.

In terms of GPU consumption, it can be observed that our proposed components exhibit superior efficiency, which seldom increases GPU memory occupancy under a similar number of voxels. Especially, this achievement is contributed by our efficiency-focused designs in constraint selection, restrained voxel-level regularizations, and combined with the efficient coding implementation. Moreover, while our proposed techniques effectively enforce a correct geometry to be learned, the artifacts, redundant voxels, and distorted surfaces can be removed or well corrected, therefore, the GPU memory requirement could even decrease along with the number of used voxels reduced.

Table 9: Efficiency Analysis of the Ablation Study.

| Items | Settings | # Voxels (M) ↓ | FPS ↑ | Peak GPU Mem ↓ | F1-Score ↑ | Training Time ↓ |
|---|---|---|---|---|---|---|
| A. | Base (SVRaster) | 9.3 | 132.9 | 12.3 GB | 0.397 | 23.8 min |
| B. | A. + Patch-wise Depth | 9.1 | 138.2 | 10.5 GB | 0.449 | 24.2 min |
| C. | B. + Multi-view Reg. | 9.1 | 138.7 | 11.4 GB | 0.538 | 65.1 min |
| | B. + Multi-view Reg. + Voxel Dropout | 9.1 | 137.3 | 11.5 GB | 0.546 | 68.3 min |
| D. | C. + Surface Rectif. | 8.8 | 146.4 | 11.1 GB | 0.549 | 64.4 min |
| | C. + Surface Rectif. + Scaling Penalty | 9.0 | 142.4 | 11.2 GB | 0.552 | 67.3 min |
| E. | D. + Voxel-Uncertainty Depth (Ours) | 8.8 | 142.0 | 11.2 GB | 0.560 | 67.5 min |

## G Additional Visualization Results

Here we show the additional visualization of the reconstructed meshes on the three datasets. In Figure 11 and 12, we show the reconstructed scenes in the TnT and Mip-NeRF 360 datasets. Our proposed GeoSVR reconstructs high-quality meshes in these complex and intricate scenes. In Figure 13, we reported the reconstructed objects in DTU datasets, and additionally show the colored rendering in Figure 14. These results prove the capability of GeoSVR to reconstruct vivid objects with accurate detail, which further proves its practical value in real-world applications. For intuitive comparison, we additionally provide a supplementary video and kindly invite the reader to watch.

## H Societal Impacts

Our proposed method provides high-quality surface reconstruction from images. So far, we have not discovered the direct negative societal impact, but it's notable that the accurate 3D reconstructions may be used maliciously. And the accurate reconstructions from real-world data may raise potential privacy concerns. During use, these societal impacts should be treated with caution.

## I Discussion

In this work, we present GeoSVR to achieve high-quality surface reconstruction with state-of-the-art accuracy, completeness, and detail preservation. Moreover, our investigation goes a further step for the possibility of recovering accurate geometry via the voxel-based representation, and also reveals a potential to better utilize the growingly important and well-established geometric foundation models [68, 69, 2, 61, 30, 4] besides the Gaussian Splatting-based approaches [57, 36, 14, 35, 15].

Nevertheless, the limitations still exist, mainly in 1) regions with serious reflections, 2) textureless areas, and 3) transparent surfaces. Due to the heavy misleading of the photometric inconsistency and the limited representation capability for ray tracing, these regions often cause suboptimal geometry.

Typically, it's a common phenomenon that the rendering quality slightly drops when the model is forced to learn the accurate geometry. This mainly lies in that the captured multi-view images from

the real world do not always retain an ideal photometric multi-view consistency that matches the correct geometry, such as the reflective regions and transparent materials, especially the current radiance fields for surface reconstruction seldom consider complex ray tracing, but mostly only once forward. In that situation, a distorted geometry in a local minimum may be better than the correct one to express an approximately accurate appearance.

In the future, introducing more efficient ray tracing techniques [23, 44, 63, 24, 7], improved voxel globality, and solutions for transparency [37, 29] could be of help to solve these challenging issues.

