# OpenReview forum: "GeoSVR: Taming Sparse Voxels for Geometrically Accurate Surface Reconstruction"
_NeurIPS.cc/2025/Conference — NeurIPS 2025 spotlight_

### Official Review · Reviewer_ZuxM · 2025-06-24

**Clarity:** 2
**Significance:** 3
**Originality:** 3
**Rating:** 5
**Confidence:** 4

**Summary:**

This paper proposes GeoSVR, a method that explores accurate surface reconstruction by building upon Sparse Voxels Rasterization aka SVRaster (Sun et al, CVPR 2025). SVRaster is similar to 3D Gaussian Splatting but encodes colors and densities within a sparse voxel grid instead of anisotropic 3D Gaussians.

GeoSVR improves SVRaster's surface reconstruction via separate regularizers, most notably a patch-wise depth/normal regularizer based on a pretrained monocular depth estimator (DepthAnythingV2). A challenge of this strategy is how to best handle inaccurate predictions from the depth estimator, which the paper suggests solving with downscaling the regularization loss with pixel-wise "uncertainty". The uncertainty score is calculated as a function of the voxel sizes used to volume-render the given pixel - the intuition is that since SVRaster progressively divides voxels at level l with large gradient to level l+1, voxels at lower levels are less constrained / should have higher uncertainty. GeoSVR also uses voxel dropout regularization to encourage a better global structure, a "surface rectification" term that encourages sharp boundaries between surfaces and empty spaces, and a scaling penalty that penalizes voxels occupying long sampling distances.

GeoSVR is evaluated against commonly used implicit benchmarks (VolSDF, NeuS, etc), and explicit methods (2DGS, SVRaster, GOF, etc). On DTU, it generally outperforms baselines with regards to Chamfer distance and on Tanks and Temples with regards to F1 score. The results for appearance quality on MipNeRF 360 are more mixed (best PSNR on outdoor scenes, 0.5db worse on PSNR than 3DGS and >1db than Mip-NeRF 360 on indoor scenes) but generally competitive.

**Questions:**

I currently feel positive about the paper, and am happy to maintain/increase my rating if the authors can address my concerns about the evaluation section. The other outstanding question I have is what GeoSVR's rendering speed looks like compared to SVRaster and 3DGS.

Typos:

Line 72-73: "An exploration GeoSVR to build explicit voxel-based framework for accurate surface reconstruction, taming sparse voxels to enable delicate and complete geometry learning."
Line 187-188: "To ensure adaptive and robust for various stages and scenarios, we obtain statistics of L to set the hyperparameters in Eq. (4)."
Line 235: "renderng" -> "rendering"

**Ethical Concerns:**

["NO or VERY MINOR ethics concerns only"]

**Final Justification:**

Thank you for answering my questions. I remain positive about the paper and will maintain my accept rating.

**Limitations:**

Right now the paper tangentially mentions "conquering challenges like varying lights and textureless regions" in the conclusion without explicitly calling out limitations of the current method. I recommend expanding upon this in an explicit limitations section along with a discussion of why appearance does not quite match existing SOTA in several cases.

**Quality:**

3

**Strengths And Weaknesses:**

One of the core ideas behind this paper (using patch-wise depth downweighed by uncertainty) seems like a clear, easy-to-understand, and effective idea, and I like how simple it is to define "uncertainty" as a function of voxel size. Some of the other regularization terms (namely the "Geometry Regularization with Voxel Dropout") are less intuitive to me, but seem to help as shown in the ablation.

The evaluation section could be improved. The metrics in Tables 1 and 2 are different and would feel less "cherry-picked" if both Chamfer Distance and F1 Score were reported in both cases. The visual metrics (PSNR/SSIM/LPIPS) should also be reported for DTU and TnT as the takeaways from the MipNeRF 360 dataset are unclear. More discussion around the appearance reconstruction results (beyond merely stating that GeoSVR is "competitive") and why it performs worse on indoor scenes relative to 3DGS/M360 would strengthen the paper.

Also, as the paper emphasized how 3DGS methods are constrained by inaccuracy of SfM initialization, it would be worthwhile to compare to MCMC ("3D Gaussian Splatting as Markov Chain Monte Carlo", NeurIPS 2024) as a popular 3DGS variant that claims competitive results without initialization.

---

> ### Author Rebuttal · Authors · 2025-07-29
>
> ### **Responses to Reviewer ZuxM**
>
> We sincerely appreciate Reviewer ZuxM's time and efforts for the positive comments and thorough suggestions. We notice that the concerns mainly relate to the metrics selection across datasets, analysis about the rendering quality, comparison with the related approach, and the running overhead. Here, we address the questions raised by the reviewer individually:
>
> ------
>
> > **Q1: Metric consistency across DTU, TnT and Mip-NeRF 360 datasets.**
>
> > - The selection of Chamfer Distance and F1-Score on DTU and TnT datasets.
>
> We agree with the reviewer that adding more comprehensive and consistent metric evaluation across different datasets is extremely important for the community, but it may be particularly difficult to implement currently. In our work, we separately adopt F1-Score and Chamfer Distance for TnT and DTU datasets **following the standard settings in previous works with no manual selection** to produce fair and persuasive comparisons. We'd like to note that these two dataset-metric bindings are generally considered as **the prevailing standard settings for surface reconstruction currently**, which have been widely used and accepted by the famous works like VolSDF, NeuralWarp, NeuS, Neuralangelo, 2DGS and etc. for years. Especially, F1-Score is the officially specified metric for the TnT dataset equipped with an official evaluation toolkit; meanwhile, using Chamfer Distance to evaluate DTU datasets with the officially provided distance calculation process has also been a well-recognized tradition in the field of surface reconstruction.  Although a unified metric setting would be better, ideally, designing and implementing a fair, objective, and robust evaluation tool of a new metric for a specific dataset from scratch indeed requires a lot of effort and time, which may be out of the scope of this paper.
>
> We fully understand the concern from the reviewer and sincerely appreciate the provided thoughtful suggestions. Currently, considering exhibiting strong performance in the most prevailing and well-accepted standard settings across the two mainstream and challenging datasets TnT and DTU, combined with the rendering quality evaluation on Mip-NeRF 360 and various reported visualization results, **we believe the contribution of our work can be well verified and recognized by the provided fair, comprehensive and objective supports in the paper.**
>
> > - Additional rendering performance on DTU and TnT datasets.
>
> Thanks for your constructive suggestion and thorough consideration. We'd like to express that evaluating rendering performance on the TnT and DTU datasets with the surface reconstruction task is not common in the field, mainly because these reconstruction settings are different from the novel view synthesis task (like the one applied to Mip-NeRF 360 setting), of which **the formers use all the captured views as the inputs in the training to reconstruct a surface model for evaluation, therefore there's not an unseen test set of the RGB views left for a fair and objective evaluation**. Considering a potential value of justification, following the comments, we provide a comparison here that conducts the rendering quality evaluation on the training set. But we want to note that these results should be interpreted with caution because these data have not been isolated during the training. Therefore, they can not fairly and justly reflect the performance of each method. We sincerely recommend referring to the results in the Mip-NeRF 360 setting for a fair, objective, and persuasive comparison.
>
>
>  *Table R1. Rendering Quality on DTU and TnT datasets. **(The evaluation can be only conducted on the training set as no test set is available, and should be interpreted with caution.)***
>
> | **Methods** | TnT||| DTU|||
> | ----- | ------ | ----- | ------ | ------ | ------ | ----- |
> || PSNR $\uparrow$ (Training Set) | LPIPS $\downarrow$ (Training Set) | SSIM $\uparrow$ (Training Set) | PSNR $\uparrow$ (Training Set) | LPIPS $\downarrow$ (Training Set) | SSIM $\uparrow$ (Training Set) |
> | SVRaster    | 24.68| 0.149| 0.866| 35.13| 0.092| 0.955|
> | 2DGS| 25.26| 0.200| 0.850| 34.32| 0.168| 0.938|
> | PGSR| 24.57| 0.150| 0.872| 33.78| 0.152| 0.945|
> | GeoSVR| 23.49| 0.150| 0.853| 32.49| 0.103| 0.949|
>
> ------
>
> > **Q2: Analysis on the novel view synthesis performance.**
>
> Thanks for the constructive suggestion! While achieving the core target of high-quality surface reconstruction, the experiments of novel view synthesis in Mip-NeRF 360 datasets show that our method also maintains a strong capability in appearance reconstruction, striking a great balance between these two tasks. Typically, it's a common phenomenon that the rendering quality slightly drops when the model is forced to learn the accurate geometry (e.g., SuGaR, 2DGS, VCR-GauS, PGSR compared to 3DGS). This mainly lies in that the captured multi-view images from the real world do not always retain an ideal photometric multi-view consistency that matches the correct geometry, such as the reflective regions and transparent materials, especially the current radiance fields for surface reconstruction seldom consider complex ray tracing, but mostly only once forward. In that situation, **a distorted geometry in a local minimum may be better than the correct one to express an approximately accurate appearance.** Especially, in Mip-NeRF 360, these kinds of regions are **more common in the Indoor Scenes**, like the metal and glasses in Counter, a large reflective television screen in Room, etc. Therefore, our performance in the Indoor categorization is worse than that in Outdoor, and may also **perform lower rendering quality than the baseline methods with relatively less accuracy in geometry**.
>
> Following the suggestion, these discussions will be added in the revised version, and we'll explore achieving better consistency between the accurate geometry and challenging appearance cases in the future.
>
> ------
>
> > **Q3: Comparison with 3DGS-MCMC.**
>
> Following the suggestion, we provide an additional comparison with 3DGS-MCMC here, where **GeoSVR still achieves SOTA performance, which further enhances the advancement and significance of our GeoSVR to explore and achieve high-quality surface reconstruction on the under-investigated representation**. 3DGS-MCMC makes important progress by reformulating 3DGS as MCMC sampling for optimization, improving the rendering quality and robustness of initialization for novel view synthesis. Considering it was not designed for surface reconstruction initially, we made the evaluation by applying it to the SOTA 3DGS baseline PGSR for fairness. As shown in Table R2, while 3DGS-MCMC relieves the reliance on initial SfM point clouds by enabling random initialization, our method still performs the best in the comparison. Due to the different focus of the tasks, some designs in 3DGS-MCMC, like the noise-driven SGLD, can bring significant improvements in appearance fitting, but may not quite match the goal of accurate surface formation currently.
>
> *Table R2. Comparisons with 3DGS-MCMC on TnT dataset. ("SfM" and "Random" denote the initialization type, and "Depth" indicates that Patch-wise Depth loss is adopted. We marked the changes of replacing the initialization from "SfM" to "Random".)*
>
> || Barn| Caterpillar| Courthouse | Ignatius| Meetingroom| Truck|
> | ---- | ---- | ----- | ---- | ---- | ---- | ---- |
> | PGSR (SfM)| 0.655| 0.486| 0.238| 0.811| 0.319| 0.652|
> | 3DGS-MCMC + PGSR (SfM)| 0.633| 0.406| *Unconverged*| 0.802| 0.315| 0.632|
> | 3DGS-MCMC + PGSR (Random)| 0.633 (+0.000) | 0.392 (-0.014) | *Unconverged*| 0.802 (+0.000) | 0.326 (+0.011) | 0.625 (-0.007) |
> | 3DGS-MCMC + PGSR (SfM+Depth)| 0.666| 0.422| *Unconverged*| 0.745| 0.327| 0.633|
> | 3DGS-MCMC + PGSR (Random+Depth) | 0.655 (-0.011) | 0.397 (-0.025) | *Unconverged*| 0.767 (+0.022) | 0.324 (-0.003) | 0.600 (-0.033) |
> | **GeoSVR (Ours)**| **0.681**| **0.494**| **0.340**  | **0.826**| **0.368**| **0.655**|
>
> ------
>
> > **Q4: Rendering speed comparison.**
>
> Our method **keeps a high efficiency in the rendering FPS, outperforming the 3DGS baselines in most cases**, and is even slightly faster than SVRaster in practice. In Section E of the supplementary material, we briefly discussed the rendering speed of GeoSVR and its relation to SVRaster. As analyzed, since our method does not introduce any burden during inference, its rendering speed is theoretically the same as SVRaster under the same number of voxels. Here we additionally compare our method with SVRaster and the 3DGS baselines on the three datasets in Table R3, and report the inference speed of our ablation study in Table R4. These results have shown the superior inference speed of GeoSVR compared to various baselines. And by forcing an accurate geometry to be learned, the voxels can be utilized more efficiently by removing distorted surfaces and artifacts, which further improves the advantage of GeoSVR in terms of efficiency.
>
>
> *Table R3. Comparisons on Inference FPS.*
>
> | Methods    | Mip-NeRF 360  | DTU| Tanks and Temples   |
> | ---- | :----: | :----: | :---: |
> | 2DGS| 25.8 | 114.9 | 88.6  |
> | GOF| 9.2  | 20.8  | 15.8  |
> | PGSR| 40.1 | 162.6 | 54.5  |
> | SVRaster   | 82.7 | 114.4 | 132.9 |
> | **GeoSVR** | 83.1 | 143.8 | 142.0 |
>
>
> *Table R4. Comparisons on Inference and Number of Voxels on TnT dataset.*
>
> | Items | Settings| # Voxels (Million) | Inference FPS |
> | ----- | ---- | ----- | ---- |
> | A.| Base (SVRaster)| 9.3| 132.9|
> | B.| A. + Patch-wise Depth| 9.1| 138.2|
> | C.| B. + Multi-view Reg. + Voxel Dropout| 9.1| 137.3|
> | D.| C. + Surface Rectif. + Scaling Penalty  | 9.0| 142.4|
> | E.| D. + Voxel-Uncertainty Depth (**Ours**) | 8.8| 142.0|
>
> ------
>
> > **Q5: Writing problems.**
>
> Thanks for your careful notice and pointing out the problems! We have fixed these errors and checked the paper thoroughly to ensure writing quality. The revisions will be included at the next opportunity to upload the paper.

---

### Official Review · Reviewer_dj1z · 2025-06-30

**Clarity:** 3
**Significance:** 3
**Originality:** 2
**Rating:** 4
**Confidence:** 4

**Summary:**

This paper proposes a method for improving sparse voxel-based surface reconstruction. Building upon the existing work Sparse Voxels Rasterization (SVR), it introduces two key enhancements: (1) a voxel-uncertainty depth constraint to leverage monocular depth cues, and (2) a surface regularization module to improve geometric consistency. Experiments on various datasets demonstrate the effectiveness of these improvements.

**Questions:**

Please see Weakness.

My main concern is Table 4 indicates most improvement stems from Multi-view Regularization [7] and Patch-wise depth [27], casting doubt on the novelty of the other components.

**Ethical Concerns:**

["NO or VERY MINOR ethics concerns only"]

**Limitations:**

Yes, no problems here.

**Paper Formatting Concerns:**

None.

**Quality:**

3

**Strengths And Weaknesses:**

Strengths:
* The work is well-motivated and builds naturally on recent developments in the community.
* The proposed method is clearly positioned within existing literature, and experiments across multiple datasets demonstrate its effectiveness.

Weakness:
* The proposed method integrates five distinct loss components. While Table 4 provides quantitative ablation results, Figures 6 and 7 only visualize the effects of two main components. As a result, the qualitative impact of the remaining loss terms remains unclear. Including more targeted visualizations—particularly for the lesser-emphasized losses—would better illustrate their contribution and justify their inclusion. Additionally, Figure 7 does not clearly demonstrate improvements in surface sharpness.

* According to Table 4, most performance gains are attributed to the Multi-view Regularization module (borrowed from [7]), which raises concerns about the novelty and technical contribution of the remaining components. A deeper analysis or improved visualization would strengthen the case for the proposed method’s overall contribution.

* The proposed method introduces additional components that improve rendering quality, but it is reported to be roughly 10× slower than the baseline SVRaster. However, this significant runtime overhead is not clearly reflected in Table 4. I suggest including time comparisons in Table 4.

---

> ### Author Rebuttal · Authors · 2025-07-29
>
> ### **Responses to Reviewer dj1z**
>
> We sincerely thank Reviewer dj1z for the time and effort in the constructive and encouraging review. We found the reviewer’s main concerns are related to how the contributions of various loss terms and runtime overhead. Here, we address the specific questions from the reviewer individually:
>
> ------
>
>
>
> > **Q1: Reported visualized effects of only two components in Figures 6 and 7. It's better to provide more visualizations.**
>
> Thanks for the constructive suggestion! In our submission, we visualized the ablation results for the three most insightful components in **Figures 6 and 7** of the main paper, and **Figures 9 and 10** of the supplementary material, along with detailed analysis in **Section A.1 and Section A.2**. We believe these visualizations can provide comprehensive support for justifying the effects of Voxel-Uncertainty Depth Constraint,  Voxel Dropout, and Surface Rectification.
>
>
> Specifically, **1)** for the Voxel-Uncertainty Depth Constraint, Figure 6 in the main paper and Figure 9 in the supplementary material have shown an obvious improvement is brought in constraint the scene geometry; **2)** For the Voxel Dropout, due to the space limitation, we reported the visualization in the Figure 10 in the supplementary material and made analysis in the Section A.2; **3)** For the Surface Rectification, some visualizations were reported in the Figure 7, just as noticed by the reviewer. Especially, we'd like to briefly explain here that our Surface Rectification is designed for **pursuing the challenging target of extremely high-precision surfaces** reconstruction. It works mainly in the stages of an almost well-reconstructed geometry to certify the delicate and difficult details, and thus affects mainly in the details that is not as obvious in visualization as the other techniques that aims to reshape a coarsely correct geometry from a worse start, but this does not equals a lower importance, **especially considering the raising difficulty and criticalness in accuracy improvement the situation that the reconstruction results have already been very close to the ground truths**.
>
> We agree with the reviewer that more visualization results would be beneficial to justify the inclusion of the components, and will add the visualized comparisons for the remaining Scaling Penalty technique in the revised version. Due to the official restriction of the rebuttal in forbidding images, we are sorry that the materials can not be presented immediately, but will be included along with the next chance of paper update.
>
> ------
>
>
>
> > **Q2: Performance Gaining of Multi-view Regularization and Patch-wise Depth.**
>
> > - Explanation and analysis of the performance in the ablation study.
>
> Thanks for your thoughtful feedback! We'd like to clarify that Multi-view Regularization and Patch-wise Depth **did not provide the key improvement in performance**. From the ablation study in Table 4, it's notable that although Multi-view Regularization and Patch-wise Depth work well, the improvements are based on a weak initial performance and have achieved an upper-bound of 0.538 in F1-Score, which is limited by the insufficient capacity to address the valuable problems our paper has revealed and aimed to solve. Notably, this score is only comparable to the previous SOTA method PGSR with depth prior added, **which has not shown the advancement that our GeoSVR achieved**. Based on this start, the results from settings of C, D, and E show that notable improvements have been brought by our proposed Voxel-Uncertainty Depth Constraint, Voxel Dropout, Surface Rectification, and Scaling Penalty, and achieve a remarkable 0.560 F1-Score. Combined with our reported visualization results, **the critical and irreplaceable effect of each component can be well justified.**
>
> Besides notable improvements have been achieved by our technical contributions, we'd like to discuss that **the progress made in different cases should be evaluated distinguishably, which correspond to different difficulties**, Especially, in the situation where GeoSVR exhibits its advancements, the scenes are already nearly well reconstructed, and the remaining problems are much more difficult and challenging to solve. In this case, **the progress we make is particularly critical and important, and is of great value for inspiring the pursuit of higher-precision reconstruction in the future.**
>
> We'd like to note that Multi-view Regularization and Patch-wise depth are not the core contributions of our work, which has been clearly described in our paper. **These two classic and general techniques are adopted to provide a basic capability to support our advanced research, but do not act as the critical innovations of our work.** We express our sincere respect for these great prior works and reported their effects in the ablation study to ensure the transparency, but would like to emphasize that the patch-warping based multi-view consistency is a long-standing effective technique for accurate geometry learning that has been widely used in many previous works for over years [18, 44, 65, 16, 12, 7, 41, 8, 47], which has become a common and standard solution for the field; and we regard the patch-wise depth just as one of an under-investigated loss function in the field of surface reconstruction for our selected monocular depth constraint, which should be well introduced in the paper.
>
> As recognized by the reviewers, **the contribution of GeoSVR mainly lies in the proposed Voxel-Uncertainty Depth Constraint, Voxel Dropout, Surface Rectification, and Scaling Penalty**. The Voxel-Uncertainty Depth Constraint, which is highly praised by the reviewers, is an insightful and sound way to maximize the effect of monocular depth cues while presenting a voxel-oriented uncertainty to avoid quality degradation. It innovatively analyzes the geometric uncertainty based on the voxel representation, enabling effective and robust scene constraints while preserving highly accurate geometries. Then, we systematically analyzed the problems in voxel-based surface formation, and targetedly proposed the Voxel Dropout, Surface Rectification, and Scaling Penalty in the aspect of multi-view geometry regularization and the bias in surface-voxel alignment. **Our investigation of high-quality sparse voxel-based surface reconstruction brings new understanding to the community.** With the thorough problem analysis and our insightful solution, GeoSVR is composed and demonstrates strong performance across various benchmarks. And besides the state-of-the-art performance achieved, **the revealed problems also bring fresh insights and could inspire more advancements to facilitate the development of the field.**
>
>
> ------
>
>
>
> > **Q3: Runtime overhead of Table 4.**
>
>
>
> | Item Notation | Settings                                | # Voxels (Million) | Inference FPS | Training Peak GPU Mem (GB) | F1-Score | Training Time |
> | ------------- | --------------------------------------- | ------------------ | ------------- | -------------------------- | -------- | ------------- |
> | A.            | Base (SVRaster)                         | 9.3                | 132.9         | 12.3                       | 0.397    | 23.8 min      |
> | B.            | A. + Patch-wise Depth                   | 9.1                | 138.2         | 10.5                       | 0.449    | 24.2 min      |
> |               | B. + Multi-view Reg.                    | 9.1                | 138.7         | 11.4                       | 0.538    | 65.1 min      |
> | C.            | B. + Multi-view Reg. + Voxel Dropout    | 9.1                | 137.3         | 11.5                       | 0.546    | 68.3 min      |
> |               | C. + Surface Rectif.                    | 8.8                | 146.4         | 11.1                       | 0.549    | 64.4 min      |
> | D.            | C. + Surface Rectif. + Scaling Penalty  | 9.0                | 142.4         | 11.2                       | 0.552    | 67.3 min      |
> | E.            | D. + Voxel-Uncertainty Depth (**Ours**) | 8.8                | 142.0         | 11.2                       | 0.560    | 67.5 min      |
>
>
>
> Thanks for the constructive suggestion! Here we provided the efficiency metrics of the ablation study Table 4. From the results, it can be observed that the multi-view regularization contributes most to the training time consumption increasing, while the other components **maintain high efficiency in all aspects, including the inference FPS, memory overheads, and the required number of voxels**. According to our analysis, this is mainly caused by the less efficient code implementation of the regularization, which we plan to solve in the future. The training time of the base SVRaster is a bit different from the official result in Table 1, which is caused by a slight hyperparameter adjustment to obtain an ideal reconstruction accuracy, which is consistent with the details we reported in lines 245-251 and is the same as the other ablation settings. We'll add the results in the revised version following the suggestion.

---

> > ### Author Response · Authors · 2025-08-07
> >
> > Dear Reviewer dj1z,
> >
> > Thank you for your constructive suggestions and thoughtful questions to help improve our paper! Here we have provided a reply to the questions in the review and hope it has clarified the key points. As the reviewer-author discussion period approaches its end, we will be grateful to receive your valuable feedback and advice, and wonder if there are any remaining questions we could address. If our response has resolved your concerns, we would greatly appreciate your consideration of a raised rating for our work.  Thank you once again for your time and effort!
> >
> > Best regards,
> >
> > GeoSVR Authors

---

> > > ### Comment · Reviewer_dj1z · 2025-08-07
> > >
> > > Dear Authors,
> > >
> > > Apologies that I could not respond earlier and thanks for the rebuttal. Yes, this rebuttal adresses most of my comments and I am happy to keep my score.
> > >
> > > Best.

---

> > > > ### Author Response · Authors · 2025-08-08
> > > >
> > > > Dear Reviewer dj1z,
> > > >
> > > > Thanks for your feedback. We've incorporated the supplemented content according to your suggestions in the revision to refine our work, and would be glad to address any further concerns you might have in the remaining discussion period. We sincerely appreciate your recognition and support in the review.
> > > >
> > > > Best regards,
> > > >
> > > > GeoSVR Authors

---

### Official Review · Reviewer_fLdW · 2025-07-02

**Clarity:** 3
**Significance:** 3
**Originality:** 2
**Rating:** 4
**Confidence:** 4

**Summary:**

This paper is a subsequent work of SVRaster, focusing on improving the geometry reconstruction quality for the voxel-based 3D representation. This task is very important and this work paves a solid step in this direction. Specifically, the author employs the prior from DepthAnything v2 to selectively guide the poorly optimized region. This idea should be generalizable across different 3D representation. Furtherly, this paper proposes some geometry-friendly regularizations, including the voxel dropout, Surface Rectification, etc.. According to the experiments, this paper outperforms others in reconstruction quality.

**Questions:**

1. Please provide more thorough analysis on the depth design.
2. Whether will the Voxel Dropout design encourage larger and more redundant voxels?

**Ethical Concerns:**

["NO or VERY MINOR ethics concerns only"]

**Final Justification:**

I keep my original score for acceptance because the rebuttal addressed most of my concerns and the proposed method is very effective.

**Limitations:**

yes.

**Paper Formatting Concerns:**

No.

**Quality:**

3

**Strengths And Weaknesses:**

Strengths:
+: Each design is simple yet effective, accompanying with a detailed analysis, making it convincing.
+: The qualitative and quantitive results outperform baselines.

Weakness:
-: The memory consumption is not clear. The author adopts some new losses, regulations, constraints, although they lead to an improvement in the geometry, it would be better to add the increasing in memory consumption to better justify.
-: The key design is the voxel-uncertainty depth guidance. While there is no direct comparison between naive depth guidance and the voxel-uncertainty version.
-: How can the method deal with the case when the depth prediction network totally fails? Where the per-scene optimization pipeline can still provides with a not bad lower bound.

---

> ### Author Rebuttal · Authors · 2025-07-29
>
> ### **Responses to Reviewer fLdW**
>
> We sincerely thank Reviewer fLdW for the positive evaluation and insightful feedback. We noticed that the reviewer’s main concerns involve analysis of computational overhead, ablation detail for depth constraint, and robustness in failure scenarios. Here, we address the questions from the reviewer individually:
>
> ------
>
>
>
> > **Q1: Memory consumption of the proposed method.**
>
> Thanks for the constructive suggestion! Our method **maintains similar or even less GPU memory overhead compared to SVRaster, and practically is able to reduce over 8% of GPU memory** (over 1.1 GB in the original 12.3 GB) in the experiments on TnT datasets on average. Here we supplement the metrics of running overheads corresponding to the ablation study of Table 4 to show the consumption of each component. From the reported results, it can be observed that our proposed components exhibit superior efficiency in GPU consumption, which **seldom increases GPU memory occupancy under a similar number of voxels**. Especially, this achievement is contributed by our efficiency-focused designs in constraint selection, restrained voxel-level regularizations, and combined with the efficient coding implementation. Moreover, while our proposed techniques effectively enforce a correct geometry to be learned, the artifacts, redundant voxels, and distorted surfaces can be removed or well corrected, **therefore, the GPU memory requirement could even decrease along with the number of used voxels reduced**. This demonstrates the high efficiency and effectiveness in improving the surface reconstruction quality in the methodology, but not solely an efficiency-quality trade-off. We'll add these results in the revised version to support a more comprehensive justification.
>
> | Item Notation | Settings                                | Inference FPS | # Voxels (Million) | Training Peak GPU Mem (GB) |
> | ------------- | --------------------------------------- | ------------- | ------------------ | -------------------------- |
> | A.            | Base                                    | 132.9         | 9.3                | 12.3                       |
> | B.            | A. + Patch-wise Depth                   | 138.2         | 9.1                | 10.5                       |
> |               | B. + Multi-view Reg.                    | 138.7         | 9.1                | 11.4                       |
> | C.            | B. + Multi-view Reg. + Voxel Dropout    | 137.3         | 9.1                | 11.5                       |
> |               | C. + Surface Rectif.                    | 146.4         | 8.8                | 11.1                       |
> | D.            | C. + Surface Rectif. + Scaling Penalty  | 142.4         | 9.0                | 11.2                       |
> | E.            | D. + Voxel-Uncertainty Depth (**Ours**) | 142.0         | 8.8                | 11.2                       |
>
>
>
> ------
>
>
>
> > **Q2&Q3: Comparison with naive depth constraint. & Robustness when the monocular depth prior fails.**
>
> Thanks for the feedback from the reviewer. From Table 4 in the main paper and Table 7 in the supplementary material, and the corresponding qualitative results in Figure 6  in the main paper and Figure 9 in the supplementary material, **our method, along with the Voxel-Uncertainty Depth Constraint, can be well verified to be effective and robust in various cases with comprehensive validations**. Here we provide our analysis.
>
> > - Q2: Comparison with naive depth constraint.
>
> In the ablation study of Table 4, we first compared the performance between the two alternatives to apply the depth priors (i.e., Sparse Points and Inverse Depth in the table), and found that our used Patch-wise Depth performs best, which is the foundation for the following ablation settings. To show the impact of our proposed Voxel-Uncertainty Depth Constraint, **we directly compared the two situations** with or without this technique in Table 4 by Item D and E. As analyzed in Section 4.2, this technique can remarkably recognize the uncertain regions and refine the geometry while preserving the well-reconstructed parts, even though the reconstruction already achieves a high quality. Recognizing the importance of a comprehensive verification, we additionally provided **a fine-grained ablation study for the depth constraint** in Section A in the supplementary material, including various comparisons on different depth estimators between no depth constraint / without Voxel-Uncertainty Depth Constraint /  with Voxel-Uncertainty Depth Constraint. According to the table, our proposed technique **consistently delivers obvious improvements in both accuracy and completeness**, and outperforms the other situations regardless of the quality of the provided depth priors, which demonstrates its effectiveness and robustness.
>
> > Q3: Robustness when the monocular depth prior fails.
>
>  Although it's extremely rare to be entirely invalid for the advanced depth estimators that are trained on large corpus, our method still support the capability for high-quality reconstruction with totally no depth priors involved, as verified in **Table 7**, which shows that **our method can still achieves a comparable performance than the previous SOTA method PGSR in such an unusual case.** Notably, in this comparison, PGSR still relies on an SfM point cloud as the prior, while our method doesn't require it. Besides this extreme case, a more common challenging situation is that the provided depth prior is of a much lower quality than expected. To verify our performance on this case, we ablate our method in Table 7 by not using the SOTA monocular estimators but the relatively old DepthAnything V1, which is much weaker in accuracy, robustness, and detail prediction than DepthAnything V2 and DepthPro, to stimulate the situation.  We find that even in this case, our method can still provide high-quality surface reconstruction, mainly thanks to our Voxel-Uncertainty Depth Constraint that successfully protects the well-reconstructed parts not influenced by the errors in the monocular depth priors, while still applying effective geometry constraints to the uncertain areas. **In summary, our method is robust for different failure cases of the depth priors.**
>
>
>
> To better guide the readers to locate the additional experiments, we'll add clearer reminders in the main paper to the related supplementary material.
>
>
>
> ------
>
>
>
> > **Q4: Will the Voxel Dropout design encourage larger and more redundant voxels?**
>
> Thanks for your insightful question! According to the efficiency ablation results from the table in the response to Q1 and observations like Figure 10 in the supplementary material, **our Voxel Dropout will not cause evidently redundant or larger voxels to be produced and harm the reconstruction quality**. This robustness can be ensured from two aspects: **1)** In our representations, the size of the voxels is strictly controlled to be unidirectional, which can only become smaller by subdivision. Therefore, the Voxel Dropout can be safely applied without the risk of distorting the voxels to fit a larger space by becoming abnormally bigger in size. **2)** In the implementation, we have separated the gradient from Voxel Dropout from the accumulated gradient indicator for voxel subdivision. As a result, the incorrect voxels that suffer from large impacts will not unexpectedly divide into a set of redundant small voxels, ensuring the sparsity control of voxels stays controllable but not worsened by the enforced regularizations. We'll add these explanations of Voxel Dropout in the revised version to help the readers better understand our method.

---

> > ### Author Response · Authors · 2025-08-07
> >
> > Dear Reviewer fLdW,
> >
> > Thank you for your valuable time and effort in reviewing our paper! We hope our reply could address your questions and concerns in the comments. As the reviewer-author discussion period is nearing its end, we would greatly appreciate your valuable feedback, and would be glad to clarify any remaining issues that might exist. If you feel our response has resolved your concerns, we would be sincerely grateful for your consideration of a raised rating.  Thank you again for your insightful comments and support!
> >
> > Best regards,
> >
> > GeoSVR Authors

---

### Official Review · Reviewer_gVE6 · 2025-07-02

**Clarity:** 2
**Significance:** 2
**Originality:** 3
**Rating:** 5
**Confidence:** 3

**Summary:**

An explicit voxel-based framework is proposed for surface reconstruction from RGB images. To better constrain surface reconstruction, a voxel-based uncertainty weighting is introduced which adaptively selects whether to rely on multi-view photometric cues or external monocular depth cues for guiding accurate surface reconstruction. To further constrain reconstruction, a multi-view geometry loss is leveraged and a form of voxel dropout is proposed when rendering depth and normals for the patch-based warping to help encourage the regularization to act more globally on the tiny voxels. Finally a surface rectification regularization is proposed for encouraging sharper surface boundaries. The proposed method builds off of SVRaster, demonstrating large improvements over SVRaster on surface reconstruction while also outperforming recent implicit and explicit surface reconstruction methods on several datasets.

**Questions:**

**Questions:**
- It seems like SVRaster needs to be run first to obtain the initial sparse voxel grid from RGB images. If SVRaster fails to provide a decent initialization will this cause the proposed approach to fail?

**Suggestions:**
- Would be good to add a description of color coding (i.e., red = best) or at least describe lower CD is better and higher F1 is better in Tables.

**Ethical Concerns:**

["NO or VERY MINOR ethics concerns only"]

**Final Justification:**

After reading the other reviews and the authors rebuttals, I have raised my score to accept. The authors addressed my major concerns in their rebuttal, and overall I think the paper offers good technical contributions while demonstrating quantitative and qualitative performance improvements over existing works.

**Limitations:**

yes

**Quality:**

3

**Strengths And Weaknesses:**

**Strengths:**
- The proposed method demonstrates strong surface reconstruction results:
    1) Significant improvement on surface reconstruction over the method the approach is based on (SVRaster), while not sacrificing performance in novel view rendering quality.
    2) Quantitatively outperforms all the other recent explicit and implicit surface reconstruction methods
    3) Qualitatively produces reconstructions that are more complete / contains less holes, results in sharper geometry, and suffers from less warped / distorted surfaces.
    4) While only slightly outperforming PGSR quantitatively, the proposed approach produces noticeably better reconstructions from the qualitative results in the main paper and supplemental.
- The proposed voxel geometric uncertainty seems like a sound way to weight whether to rely on solely photometric cues or use additional monocular depth cues, as large voxels (lower levels in octree) likely indicate regions with low overlap or low photometric consistency in images and therefore need additional supervision from depth to accurately locate surfaces. Moreover, the proposed uncertainty weighting seems to be effective regardless of foundation depth model used which further demonstrates its usefulness.

**Weaknesses:**
- It is not completely clear to me that the proposed surface rectification is an effective way to solve the issue of increasing density in one voxel causing an increased density in the neighboring voxels. While I understand the proposed surface rectification can help produce a sharper boundary, encouraging the surface to be pulled toward the "out" point seems like it may potentially shrink the geometry. Can’t this lead to undesirable artifacts or is this unlikely as the voxel sizes are typically very small? Moreover, the surface rectification seems to have very little impact quantitatively (Table 4) and qualitatively (Figure 7).
- Writing could use some improvement:
    1) Surface rectification section (lines 229 - 233) could be improved. In line 230, $\alpha_{p,e}$ and $\alpha_{p,o}$ are called voxels but then in Figure 3 $p_{e,o}$ look like crossing points between voxels, and then in Eq. 9 $\alpha_{p,e/o}$ is defined as the standard alpha term from volume rendering, which is confusing to read.
    2) Some sentences have grammatical errors (line 177, end of line 182 and beginning of 183, line 187)

---

> ### Author Rebuttal · Authors · 2025-07-29
>
> ### **Responses to Reviewer gVE6**
>
> We sincerely thank Reviewer gVE6 for recognizing our work and providing valuable comments. We noticed that the main concerns from the reviewer are the performance and explanation of the Surface Rectification design, improvements on the writing and the relation to SVRaster in initialization. Here, we address the questions individually:
>
> ------
>
>
>
> > **Q1: Robustness and Effectiveness of the Surface Rectification.**
>
> > - Will Surface Rectification cause inaccurate shrunk geometry by over-pulling the surfaces to the "out" direction?
>
> Our proposed Surface Rectification aims to address the biased surface problem at voxel level, **offering robust correction in the whole optimization process**. It can prevent the over-shrunk geometry problem in two aspects: **1)** First, while a typical effect of Surface Rectification is to solve the front-biased surface by pulling it toward the out direction, as illustrated in Figure 3, **our design does not restrict the rectification direction** but is to encourage a sharp density variation inside of one voxel by both penalizing the front density value meanwhile encouraging the after one. Therefore, in some cases, the Surface Rectification can even help fix unexpected hollows to be smooth and complete (e.g., when the density value of the enter point $\mathbf{P}_e$ has already been small but the out point  $\mathbf{P}_o$ is not large enough). The surface would not be blindly pulled towards a certain incorrect direction. **2)** As noticed by the reviewer and analyzed in our paper, an accurate surface is always formed by tiny voxels rather than the large ones. As our Surface Rectification is conducted inside specific critical voxels, its direct effect is limited to its target and the nearest neighbors. Thus, for the tiny voxels that are important to form the final accurate surfaces, **restrained yet effective rectification is adopted**, while the large voxels are still in the stage of coarser optimization with the photometric and geometric supervisions. We'll add more explanations in the revision to better clarify the working principle of Surface Rectification.
>
> > - The quantitative and qualitative effect of the Surface Rectification.
>
> Surface Rectification brings **a significant and critical improvement in high-precision surface refinement** in reconstruction accuracy, especially playing an irreplaceable role **in situations where the reconstruction results have already been very close to the ground truths**.  Different to the proposed contributions like Voxel-Uncertainty Depth that has a large and macro influence with large geometric changes to adjust the obvious errors, Surface Rectification is an important and necessary technique for **pursuing the challenging target of extremely high-precision surfaces**, which works mainly in the stages of an almost well-reconstructed geometry to certify the delicate and difficult details, such as shown in Figure 7. Especially, its effect can be well demonstrated on the DTU dataset quantitatively, as shown in Table 8 in the supplementary material. Besides the critical accuracy improvement in high-precision cases, we believe the revealed problem behind this technique can also deliver novel insights to inspire more advancements to facilitate the development of the field.
>
> Thanks for the feedback from the reviewer! To help the readers better understand, we'll add clear guidance in the main paper to the related additional experiments in the supplementary material, and provide more qualitative results for better demonstration.
>
> ------
>
>
>
> > **Q2: Writing problems.**
>
> We sincerely appreciate your careful notice to help improve our paper! In line 230, we have missed a word of "alpha" during writing, and we are sorry for causing the confusion. The correct descriptions should be "... an enter voxel alpha $a_{\textbf{p,e}}$ and the out  $a_{\textbf{p,o}}$ ...", which is calculated following Eq. (9). And following the comment, we have checked our paper thoroughly and corrected the grammatical errors, and fixed unclear expressions that may lead to misunderstanding.
>
> Specifically, for the problems pointed out by the reviewer, we have made the corrections, such as:
>
> - Line 177: "Geometric $U_{\text{geom}}(v)$ is composed of $U_{\text{base}}$ from level with voxel density, ..." -> "Geometric uncertainty $U_{\text{geom}}(v)$ is composed of the level-depended base uncertainty $U_{\text{base}}$ and the voxel density, ...";
> - Lines 182-183: "..., we resort to a patch-wise global-local depth loss for better scale alignment and facilitate the geometry knowledge learning." -> "..., we resort to a patch-wise global-local depth loss for better scale alignment and facilitating the geometry knowledge learning.";
> - Line 187: "To ensure adaptive and robust for various stages and scenarios, we obtain ..." ->  "To ensure adaptive and robust constraints for various stages and scenarios, we obtain ...".
>
> These revisions and the other unlisted writing refinements will be updated in the revised version.
>
> ------
>
>
>
> > **Q3: Relation to the initialization from vanilla SVRaster.**
>
> Indeed, **our method does not inherit reconstruction results from vanilla SVRaster**, and is able to **perform high-quality reconstruction for the challenging cases that SVRaster may fail**. We'd like to note that the pipeline of GeoSVR is not linked to the initial voxels produced by vanilla SVRaster, and is robust in situations where SVRaster can only reconstruct seriously inaccurate surfaces. As described in Section 1 and Figure 3, both our method and SVRaster adopt initialization of constant voxels that depend on the range of camera trajectory to cover the whole scene, and the reconstruction processes would go differently once the optimization starts. As shown in the first scene in the supplementary video, and comparisons of Figure 11 in the supplementary material, GeoSVR has successfully reconstructed the regions that are extremely biased from the ground truth in SVRaster, demonstrating the superior robustness of GeoSVR. Due to the restriction on the rebuttal, we are not allowed to upload more typical cases here with images, but we will add the results in the revised version to help declare the performance relation between our method and SVRaster.
>
> ------
>
>
>
> > **Q4: Better clarity of the highlighting and metrics.**
>
> Thanks for the suggestions! Following the comments, we have conducted a thorough double-check of our paper, especially on the unclear descriptions, and refined it, including adding the explanations of the color marks in Tables 1, 2, 3, and declaring the directionality of the metrics in Tables 1 and 2 in the captions. The updates will be shown in the next chance of paper uploading. We thank the reviewer again for helping us make the paper easier to read with better clarity.

---

> > ### Comment · Reviewer_gVE6 · 2025-08-03
> > **Response to authors**
> >
> > Thank you for the detailed response. My misunderstandings regarding surface rectification and the initialization from vanilla SVRaster have been addressed, therefore I have chosen to raise my score.

---

> > > ### Author Response · Authors · 2025-08-04
> > >
> > > We are grateful to hear that our response has resolved your concerns. We sincerely appreciate your thoughtful feedback and the encouraging recognition conveyed by the raised score. Once again, thanks for your valuable time and effort in reviewing our paper.
> > >
> > > Best regards,
> > >
> > > GeoSVR Authors

---

### Comment · Area_Chair_gL1U · 2025-08-05

Dear Reviewers,

Thanks for your efforts reviewing the papers. Since we have fewer than two days before the author-reviewer discussion period ends (August 6, 11:59pm AoE), could you please do the following ASAP: carefully read the other reviews and the authors' rebuttal, and reply to the authors regarding whether your questions or concerns have been addressed. If any issues remain unresolved, please specify them so that the authors have a fair opportunity to respond before the author-reviewer window closes.

If you have done so, please ignore this comment.

Best,
AC

---

### Note · Authors · 2025-08-13

We sincerely thank all reviewers and the AC for their time, effort, and constructive feedback throughout the review process. Here we summarize the discussion phase of our submission.

Initially, this paper received four consistently positive recommendations. The reviewers consider this work has made **a solid step in this important field** (fLdW) by introducing a novel sparse voxel-based framework with **interpretable and sound technical contributions** (gVE6, fLdW, ZuxM), and achieved **superior performance** (gVE6, fLdW, dj1z, ZuxM). The method contributed **clear and effective theoretical ideas** (gVE6, fLdW, ZuxM) for the advancement of high-precision surface reconstruction, delivering insights to **inspire further progress in the 3D vision community** (fLdW).

**During the discussion, we addressed all the raised questions in detail**：

- **Reviewer gVE6** initially raised questions about the details of surface rectification and the initialization, which are confirmed to be addressed by the reviewer with our clarifications.
- **Reviewer fLdW and Reviewer dj1z** suggested adding runtime overhead analysis to enhance the comprehensiveness of validation. We addressed this with additional experimental results on GPU memory, FPS and training time, and the reviewers are satisfied with the shown efficiency advantages.
-  **Reviewer dj1z** also suggested additional visualizations about the ablation study for better verification. We addressed this by the clarifications of existing results and also promising to include extra results in the revision, and the reviewer is satisfied with our response.
- **Reviewer ZuxM**’s suggestion regards extended evaluation scope with related baselines and comparison with 3DGS-MCMC for a more comprehensive verification. We addressed the queries by our clarification, providing additional experiments on rendering evaluation, FPS comparison with the baselines, and additional experiments with 3DGS-MCMC, and showed our method's further advantages. The reviewer is satisfied with the response.


Given the above combination, along with consistent recognition from all the reviewers, we believe this work makes a meaningful contribution to the field and can deliver unique highlights at NeurIPS.

---

### Decision · Program_Chairs · 2025-09-17

**Decision:**

Accept (spotlight)

**Comment:**

The final scores for this submission are 5/5/4/4. After the rebuttal period, all reviewers expressed a positive attitude toward the paper and acknowledged that the authors had addressed their main concerns. The main strengths of this work are its strong surface reconstruction performance and the simplicity and effectiveness of the proposed method. By carefully reviewing both the paper and the rebuttal, I recommend accepting this submission.